# Ezrin defines TSC complex activation at endosomal compartments through EGFR–AKT signaling

Giuliana Giamundo[1†], Daniela Intartaglia[1†], Eugenio Del Prete[2], Elena Polishchuk[2], Fabrizio Andreone[2], Marzia Ognibene[3], Sara Buonocore[1], Bruno Hay Mele[1], Francesco Giuseppe Salierno[2], Jlenia Monfregola[2], Dario Antonini[1], Paolo Grumati[2,4], Alessandra Eva[5], Rossella De Cegli[2], Ivan Conte[1*]

[1]Department of Biology, University of Naples Federico II, Naples, Italy; [2]Telethon Institute of Genetics and Medicine, Pozzuoli, Italy; [3]U.O.C. Genetica Medica, IRCCS Istituto Giannina Gaslini, Genova, Italy; [4]Clinical Medicine and Surgery, University of Naples Federico II, Naples, Italy; [5]Laboratory of Molecular Biology, IRCCS Istituto Giannina Gaslini, Genova, Italy

**\*For correspondence:**
ivan.conte@unina.it

[†]These authors contributed equally to this work

**Competing interest:** The authors declare that no competing interests exist.

## eLife Assessment

Giamundo et al. present **fundamental** data with new insights into the role of Ezrin, a major membrane-actin linker that assembles signaling complexes, in the spatial regulation of EGF signaling mediators. The use of multiple state-of-the-art microscopy techniques, multiple cell lines and inhibitors, together with in vivo models **convincingly** supports the majority of their conclusions. The findings are helpful for understanding EGF/mTOR signal transduction and support a critical role for the scaffolding protein Ezrin in the upstream regulation of EGFR/AKT activity, TSC subcellular localization and mTORC1 signaling. These findings contribute substantially to understanding how endo-lysosomal signaling are regulated, alterations which are implicated in many human diseases.

**Abstract** Endosomes have emerged as major signaling hubs where different internalized ligand–receptor complexes are integrated and the outcome of signaling pathways are organized to regulate the strength and specificity of signal transduction events. Ezrin, a major membrane–actin linker that assembles and coordinates macromolecular signaling complexes at membranes, has emerged recently as an important regulator of lysosomal function. Here, we report that endosomal-localized EGFR/Ezrin complex interacts with and triggers the inhibition of the Tuberous Sclerosis Complex (TSC complex) in response to EGF stimuli. This is regulated through activation of the AKT signaling pathway. Loss of Ezrin was not sufficient to repress TSC complex by EGF and culminated in translocation of TSC complex to lysosomes triggering suppression of mTORC1 signaling. Overexpression of constitutively active EZRIN[T567D] is sufficient to relocalize TSC complex to the endosomes and reactivate mTORC1. Our findings identify EZRIN as a critical regulator of autophagy via TSC complex in response to EGF stimuli and establish the central role of early endosomal signaling in the regulation of mTORC1. Consistently, Medaka fish deficient for Ezrin exhibit defective endo-lysosomal pathway, attributable to the compromised EGFR/AKT signaling, ultimately leading to retinal degeneration. Our data identify a pivotal mechanism of endo-lysosomal signaling involving Ezrin and its associated EGFR/TSC complex, which are essential for retinal function.

## Introduction

Endosomes are intracellular membrane-bound organelles that receive, integrate, and transmit a variety of signals to intracellular compartments. Trafficking of receptors, ion channels, lipids, and other effector proteins within the endosomal vesicles provides a mechanism to either sustain intracellular signaling pathways active (*Pálfy et al., 2012*) or to downregulate signaling pathways through their degradation in lysosomes. Accordingly, altered endosomal maturation and function play a key role in the pathogenesis of a wide range of human diseases including diabetes, cancer, and neurodegenerative disorders. Therefore, new insights about endosomal signaling and understanding the molecular components restricting signaling activity to specific pathways will uncover new opportunities for pharmacological targeting of such disorders. In the retinal pigment epithelium (RPE), endosomes contribute to the diurnal clearance of phagocytosed photoreceptor outer segments (POS) that is required for RPE and photoreceptor health. This process is linked to circadian and light phase and is initiated by the scission from the plasma membrane of phagosomes containing POS, which undergo gradual fusion with endosomes and finally with lysosomes in a coordinated process termed 'maturation'. The high demand on lysosomes for the digestion and recycling of phagocytosed POS rather than for the clearance of mitochondria, oxidized proteins, and other cellular components suggests the existence of a signaling pathway that can finely coordinate lysosomal function according to needs that will not upset cellular homeostasis. A long-standing question in the field is how functional diversity within the autophagy pathway is achieved in the RPE in the dark and light phases. Inhibition of endosomal biogenesis, trafficking, and fusion is associated with impairment of lysosomal biogenesis and autophagy flux. Implicit in these findings is the idea that endosomes, by carrying signaling molecules, could serve as a signaling hub for the regulated transfer of signals to lysosomes, acting more specifically than diffusion-based signal propagation. However, how endosomes are essential for lysosomal function, and their relevant components regulating this process are still not well defined.

Ezrin (Ezr), a member of the ezrin–radixin–moesin (ERM) protein family, is mainly localized just beneath the plasma membrane around cellular protrusions and villi. Ezrin acts as a scaffolding platform to cross-link F-actin cytoskeleton with specialized membrane components (*Kawaguchi and Asano, 2022*) that are implicated in the spatiotemporal dynamics of phagosomes and endosomes. Its association with both F-actin filaments and membrane proteins is finely regulated and requires conformational activation through phosphorylation at unique (Y353, Y477, and T567) residues. The central role of Ezrin in regulating trafficking of vesicles has been described (*Cha et al., 2006*; *Tamma et al., 2005*; *Zhou et al., 2003*). Indeed, maturation of endosomes and recycling/exocytosis of their components (i.e., α1β-adrenergic receptor, NHE3, and others) require the Ezrin protein (*Barroso-González et al., 2009*; *Cha et al., 2006*; *Stanasila et al., 2006*; *Zhao et al., 2004*; *Zhou et al., 2005*). The phosphorylated active Ezrin is observed within early and late endosomes (*Parameswaran et al., 2013*). Moreover, through its active and reversible interactions with actin filaments and endosomal proteins, Ezrin organizes signal transduction. Indeed, phosphorylation of the T567 residue of EZRIN leads to its co-localization in a functional complex with NHE1, EGFR, and β1-integrin in human breast tumors, suggesting its crucial role as a scaffold protein of EGFR (*Antelmi et al., 2013*). Accordingly, Ezrin also interacts with EGFR at membranes (*Saygideğer-Kont et al., 2016*). In mammalian cells, depletion of an Ezrin-interacting protein, Vsp11, delays the delivery of EGFR to endosomes (*Chirivino et al., 2011*), thus linking the Ezrin protein network with EGFR trafficking via clathrin-coated transport vesicles. However, the mechanisms of Ezrin–EGFR interaction and its function at the endosomal compartments remain largely unexplored. Interestingly, recent findings have shown that PI3K-mediated activation of AKT upon EGF stimulation is mediated by EGFR via an early endocytic pathway (*Nishimura et al., 2015*). These findings suggest that Ezrin may be required as a protein scaffold for coordinating EGFR/AKT signaling at endosomes. Interestingly, EZR interacts with AKT in breast cancer cells (*Li et al., 2019*). In addition, the Y353-phosphorylation of Ezrin is relevant for PI3K-initiated signaling through its interaction with p85, the regulatory subunit of PI3K (*Gautreau et al., 1999*). Notably, the Ezrin–p85 complex optimizes the physiological activation of AKT, supporting a central role of Ezrin in controlling intracellular pathways in response to external cell signaling (*Gautreau et al., 1999*).

Accumulating evidence has shown that the AKT-mediated Tuberous Sclerosis Complex (TSC complex) phosphorylation is a major mechanism in triggering the activity of the GTPase Rheb (Ras homolog enriched in brain), an essential activator of mTORC1 at lysosomes. We demonstrated recently that Ezrin is a key regulator of lysosomal biogenesis and functions in RPE/retina crosstalk by

modulating TFEB nuclear translocation (*Naso et al., 2020*). Moreover, Ezrin overexpression leads to altered autophagy and an impairment of POS maturation and degradation in RPE cells (*Naso et al., 2020*). Thus, it is intriguing that Ezrin has been recently observed to be associated with lysosomes (*Poupon et al., 2003*). Furthermore, cancer cell proliferation and invasion through an activated Akt/mTORC1 pathway was linked with activation of Ezrin (*Krishnan et al., 2006*). In contrast, depletion of Ezrin was found to be associated with the repression of the mTORC1 pathway (*Wan et al., 2005*). Together, these data led us to hypothesize that Ezrin-mediated EGFR endosomal sorting and trafficking could play a central role in mTORC1 activation on lysosomes.

Here, we identify a previously undocumented function for Ezrin as a platform that is essential for the endosomal signaling network involving EGFR and AKT pathways, which provides an important insight into the spatial inactivation of the TSC complex on endosomal compartments. We show that inactivation of Ezrin is crucial to neutralize EGF-stimulated EGFR endosomal sorting and signaling from the plasma membrane with a reduction of AKT-mediated phosphorylation of TSC complex, which in turn translocates and inhibits mTORC1 on lysosomes. These results reveal an essential layer of mTORC1 regulation by Ezrin and EGFR signaling and uncover part of the paradigm of signaling from endosomes to lysosomes to coordinate the lysosomal function in the retina and other tissues. Consistent with the role of the Ezrin/EGFR/TSC complex axis in lysosomal biogenesis and function, alteration of this molecular network alters autophagy in vivo in Medaka fish, resulting in retinal degeneration. Derangement of this control mechanism may underpin human eye disorders and may be relevant as a therapeutic target to restore normal vision.

## Results

### Ezrin regulates lysosomal biogenesis

Activated Ezrin represses the autophagy pathway in the RPE (*Naso et al., 2020*), but the mechanism remains undefined. To gain insights into this, we used integrated comparative analysis by unbiased RNA-seq and high-resolution mass spectrometry-based proteomic studies on Ezrin$^{-/-}$ mouse embryonic fibroblasts (MEFs) (EZR$^{KO}$) (*Ognibene et al., 2011*). The comparison of the transcriptomics and proteomics identified 572 commonly regulated genes: 317 and 213 genes are induced and inhibited in both datasets, respectively (*Figure 1—figure supplement 1a*). Gene Ontology and Functional Annotation Clustering analyses were performed on these 530 commonly differentially expressed genes, restricting the output Cellular Compartments (CC) terms (*Supplementary file 1*, *Supplementary file 2*, and *Supplementary file 3*). We found an enriched overlap of these genes in cell compartments, including cell membrane and lysosome (*Figure 1a*, *Figure 1—figure supplement 1a, b* and *Supplementary file 1*, *Supplementary file 2*, and *Supplementary file 3*). Consistent with this, immunofluorescence analysis revealed that the EZR$^{KO}$ displayed an increased number of lysosomes, as assessed by quantification of lysotracker-fluorescent staining and Lamp1–LC3 co-localization (*Figure 1b–f*). Compared with WT MEFs, lysosomal Cathepsin B (CTSB) activity of EZR$^{KO}$ MEFs was significantly increased (*Figure 1g*). Furthermore, western blot analysis also revealed that EZR$^{KO}$ increased the expression of lysosomal markers (LAMP1, CTSD, and LC3) as well as reducing the levels of the autophagy substrate p62 and NBR1 (*Figure 1h*). To further investigate whether this autophagic induction was Ezrin dependent, we inserted a frameshift deletion of 13 nt in the coding region (exon 2) of the *EZRIN* gene in HeLa cells via CRISPR/Cas9-mediated genome editing (EZR$^{-/-}$) (*Figure 1—figure supplement 1d*). Concordantly, we found that EZR$^{-/-}$ cells have a significantly increased number of lysosomes and increased lysosomal activity (*Figure 1—figure supplement 1e–g*), indicative of augmented lysosomal biogenesis and function. Consistent with this, western blot analysis showed that autophagic flux and lysosomal markers were also increased in EZR$^{-/-}$ compared to control (*Figure 1—figure supplement 1h, i*). Notably, we found TFEB nuclear localization as a consequence of Ezrin depletion (*Figure 1—figure supplement 1j*), in line with previous results (*Naso et al., 2020*). Taken together, these results reveal the crucial roles of Ezrin in lysosomal biogenesis and function. These results are consistent with our previous report showing that the autophagy pathway is blocked by Ezrin overexpression in vivo (*Naso et al., 2020*).

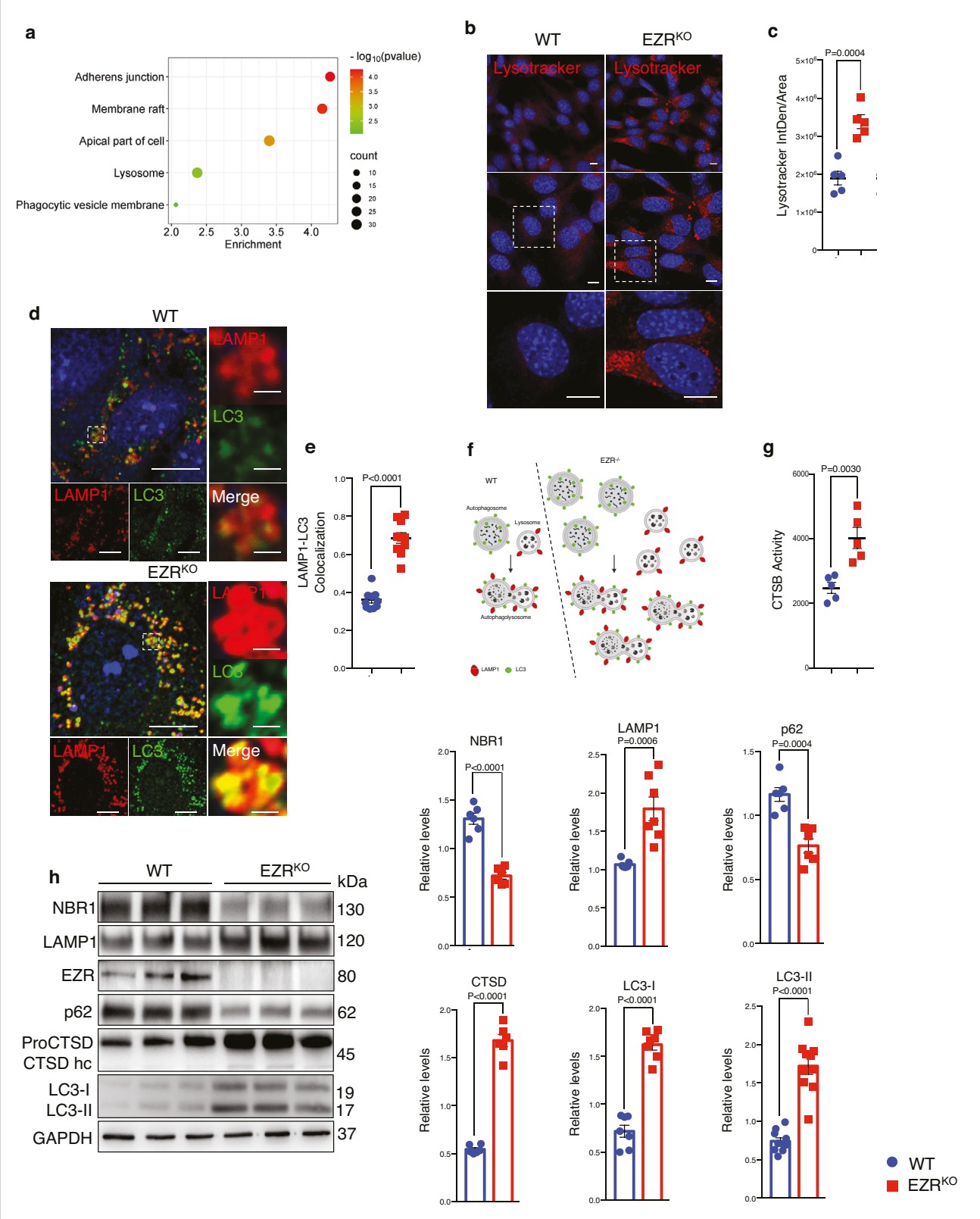

**Figure 1.** Deletion of Ezrin increases the lysosomal pathway. (**a**) Gene Ontology (GO) of 530 genes differentially expressed at mRNA and protein levels (EZR[KO] versus WT). Bubble plot representing some of the most enriched GO terms regarding cellular components. Color and *x* axis represent minus logarithms of p-value. The size represents the numbers of genes enriched in a GO term. (**b**) WT and EZR[KO] mouse embryonic fibroblast (MEF) cells were cultured in 6-cm cell plates for 24 hr, then fixed and immunostained with lysotracker and DAPI. Scale bar: 10 μm. (**c**) Data represent mean of lysotracker-

*Figure 1 continued on next page*

Figure 1 continued

positive cells ± SEM (n = 3 experiments at least). Statistical test: unpaired t-test. (**d**) MEF cells WT and Ezr$^{KO}$ were cultured in 6-well plates for 24 hr, then fixed and immunostained with LAMP1 and LC3 antibodies and DAPI. Scale bar: 10 µm (magnification 1 µm). (**e**) Data represent mean of LAMP1–LC3 co-localization spots ± SEM (n = 3 experiments at least). Statistical test: unpaired t-test. (**f**) Model showing autophagic flux induction in EZR$^{-/-}$ cells. This panel was created using BioRender.com. (**g**) MEF Ezr$^{KO}$ showed CTSB enzymatic activity increase compared to control cells. (**h**) MEF cells WT and Ezr$^{KO}$ were lysed and immunoblotted with NBR1, LAMP1, EZR, P62, and LC3 antibodies or GAPDH antibodies as a loading control. The graphs show the mean NBR1, LAMP1, EZR, P62, and LC3 levels relative to GAPDH ± SEM (n = 3 experiments at least). Statistical test: unpaired t-test for NBR1, P62, and LC3-I; Welch's t-test for CTSD and LC3-II; Mann–Whitney test for LAMP1.

The online version of this article includes the following source data and figure supplement(s) for figure 1:

**Source data 1.** Raw uncropped and unedited blots relating to *Figure 1*.

**Source data 2.** Uncropped blots with the relevant bands labeled relating to *Figure 1*.

**Figure supplement 1.** Ezrin knockout cells show lysosomal enhancement.

**Figure supplement 1—source data 1.** Raw uncropped and unedited blots relating to *Figure 1—figure supplement 1*.

**Figure supplement 1—source data 2.** Uncropped blots with the relevant bands labeled relating to *Figure 1—figure supplement 1*.

## Ezrin interacts with EGFR and regulates its activation

Previous studies have implicated Ezrin in coordinating signaling complexes on membranes in cancer, raising the question of whether the Ezrin-mediated control of autophagy may be attributed to an alteration of signaling pathways. To identify potential signaling pathways affected by Ezrin modulation, we performed an enrichment analysis of the 530 differentially expressed genes in EZR$^{KO}$, using stable isotope labeling by amino acids in cell culture (SILAC) phosphoproteomics, kinase perturbations from GEO database, and the Proteomics drug atlas. Interestingly, SILAC phosphoproteomics data highlighted a significant overlap with phosphorylation changes in HeLa cells upon EGF treatment (*Figure 2a* and *Supplementary file 4*). Accordingly, kinase perturbation revealed a significant overlap with downregulated genes upon EGFR drug activation (*Figure 2—figure supplement 1a* and *Supplementary file 4*), whereas Proteomics drug atlas revealed a significant enrichment in cells upon AZ628 (a Raf inhibitor) or MEK162 (a MEK inhibitor) (*Figure 2—figure supplement 1b* and *Supplementary file 4*; *Liu et al., 2006*; *Mitchell et al., 2023*; *Olsen et al., 2006*; *Ong et al., 2002*; *Warde-Farley et al., 2010*) (all resources are available https://maayanlab.cloud/enrichr-kg/downloads). Thus, we hypothesized that the EGFR needs to be selectively recognized by EZRIN to be subjected to EZRIN-mediated endosomal trafficking and signaling. Consistent with this, gene network based on physical interaction reveals EGFR as a possible direct EZRIN protein partner (*Figure 2b* and *Supplementary file 5*). Moreover, EGFR resulted strongly upregulated in our omics dataset (*Figure 2c*) and co-immunoprecipitation (CoIP) experiments revealed a complex composed of EZRIN and EGFR (*Figure 2d*), consistent with human biomedical interaction repositories (*Oughtred et al., 2021*; *Petschnigg et al., 2014*; *Salokas et al., 2022*). To further detail this possible interaction, we assessed all possible pairwise interactions between the known domains of Ezrin UID:P15311 and EGFR (UID:P00533) by mining 3did (10.1093/nar/gkt887) and PPIDomainMiner (10.1371/journal.pcbi.1008844) (*Figure 2—figure supplement 1c*). Interestingly, the EZRIN–EGFR interaction appears to occur between the FERM central domain (PF00373) of Ezrin with the PK domain (PF07714) of EGFR, in line with previous structural studies by X-ray diffraction in which was patterned a direct interaction of the FERM domain with kinase domain of focal adhesion kinase (FAK) (*Lietha et al., 2007*). Phosphorylated Ezrin (Thr567) localizes at curved cytoplasmic membranes and has been implicated as a membrane–cytoskeleton scaffolding protein rather than a membrane shaper (*Tsai et al., 2018*). We therefore tested whether phosphorylation at Thr567 of Ezrin was involved in interacting with EGFR in a complex at cytoplasmic membrane. As hypothesized, phosphomimic active EZR$^{T567D}$, but not phospho-mutant inactive EZR$^{T567A}$ protein (*Naso et al., 2020*), when expressed in EZR$^{-/-}$ cells, highly co-immunoprecipitated with EGFR (*Figure 2e*). These data support that active EZRIN protein interacts with EGFR. Co-immunostaining analysis confirmed that EZRIN is localized at the plasma membrane with EGFR (*Figure 2f*). Moreover, a fraction of the EZRIN signal co-localizes with EGFR, within same intracellular compartments (*Figure 2f*), supporting the presence of an EZRIN/EGFR complex, in which EZRIN acts as a scaffold protein for EGFR. Thus, we postulated that EZRIN participates in EGFR trafficking and signaling. To test this hypothesis, we examined the expression levels and subcellular distribution of EGFR under normal and EZRIN-depleted conditions by immunofluorescence staining

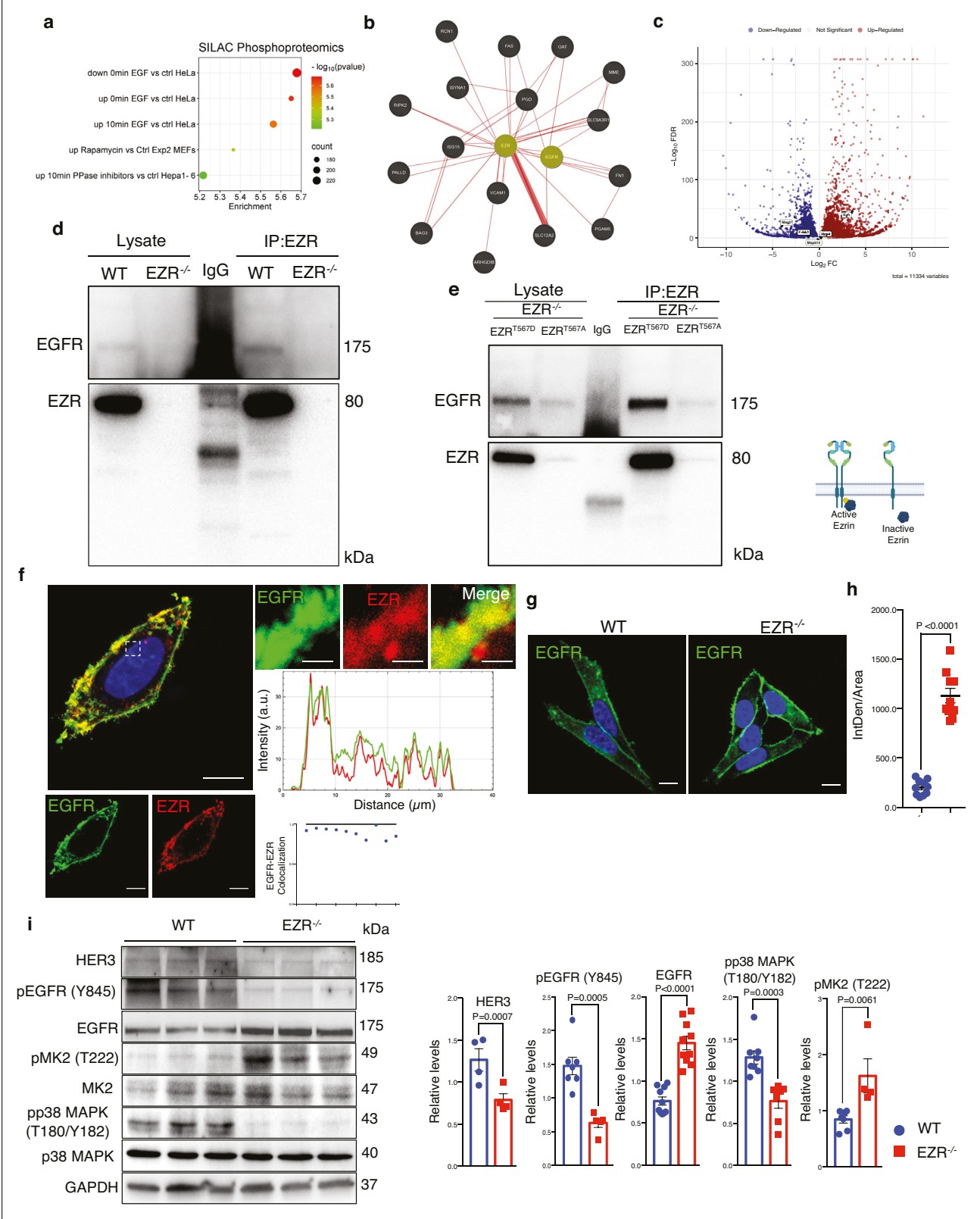

**Figure 2.** Ezrin binds EGFR and regulates its activation. (**a**) Bubble plot representing the enrichment analysis of 530 differentially expressed genes (DEGs) performed in SILAC Phosphoproteomics data. Color and x axis represent minus logarithms of p-value. The size represents numbers of genes enriched in the indicated data. (**b**) Physical interactions, obtained by GeneMANIA, highlight Ezrin and EGFR binding. (**c**) Volcano plot of DEGs, with upregulated EGFR and downregulated MAP2 and ERBB2 (no threshold on Log2FC and 0.05 threshold on −Log10FDR). Legend: red dot, upregulated

*Figure 2 continued on next page*

*Figure 2 continued*

gene; blue dot, downregulated gene; gray dot, not significant gene. (**d, e**) Co-IP data for Ezrin–EGFR interaction. For the co-IP analyses, was used Ezrin antibody, conjugated with beads, and immunoblotted with EGFR antibody for WT and EZR$^{-/-}$ (**d**) and HeLa EZR$^{T567D}$ and EZR$^{T567A}$ (**e**) HeLa cells, respectively. Schematic representation of HeLa EZR$^{T567D}$ and EZR$^{T567A}$ co-IP (bottom). This shematic was created using BioRender.com. (**f**) Confocal microscopy images showing EGFR (green) and EZR (red) co-localization on the membrane in HeLa WT cells (left) and magnified views of the regions are provided (right). Scale bar: 10 µm (magnification 1 µm). Representative plots of co-localization profiles on the membrane between EGFR (green) and EZR (red). Data represent mean of EGFR–EZR co-localization spots ± SEM (*n* = 3 experiments at least); (**g**) immunofluorescent labeling images of EGFR in HeLa WT and EZR$^{-/-}$ cells, observed by confocal microscopy. Scale bar: 10 µm. (**h**) Data represent fluorescence intensity ± SEM (*n* = 3 experiments at least). Statistical test: unpaired *t*-test; (**i**) immunoblots and calculated levels (bottom) of HER3, pY845 EGFR, EGFR, pT222 MK2, MK2, pT180/pY182 p38 MAPK, and P38 MAPK in HeLa WT and EZR$^{-/-}$ cells. Data are expressed as mean of pY845EGFR/EGFR, pT222 MK2/MK2, and pT180/pY182 p38 MAPK/ P38 MAPK ratio ± SEM (*n* = 3 experiments at least). GAPDH was used as loading control. Statistical test: unpaired *t*-test for pY845 EGFR; Mann–Whitney test for HER3, EGFR, pT222 MK2, and pT180/pY182 p38 MAPK.

The online version of this article includes the following source data and figure supplement(s) for figure 2:

**Source data 1.** Raw uncropped and unedited blots relating to *Figure 2*.

**Source data 2.** Uncropped blots with the relevant bands labeled relating to *Figure 2*.

**Figure supplement 1.** Ezrin genetic and pharmacological depletion causes EGFR signaling alteration.

**Figure supplement 1—source data 1.** Raw uncropped and unedited blots relating to *Figure 2—figure supplement 1*.

**Figure supplement 1—source data 2.** Uncropped blots with the relevant bands labeled relating to *Figure 2—figure supplement 1*.

and western blot (*Figure 2g–i*). Interestingly, the genetic depletion of EZRIN strongly induced a statistically significant localization of EGFR at the plasma membrane and dramatically reduced its presence within intracellular compartments (*Figure 2g, h*). Considering that the specific EGFR signals can arise from intracellular compartments, such as the endosomal compartment (*Burke et al., 2001*), we examined whether EZRIN depletion would impair EGFR signaling. Consistent with proteomic results, we found that the levels of total EGFR are increased in EZR$^{-/-}$ compared to control cells (*Figure 2i*). However, western blot analysis demonstrated that the absence of EZRIN induced a reduction in EGFR signaling. Notably, the level of HER3 and active pY845 EGFR were almost abolished following EZRIN depletion, in line with previous results where disruption of wild-type EGFR signaling induced reduction of HER3 (*Liu et al., 2020*). Interestingly, lack of EZRIN also reduced the EGFR-stimulated phosphorylation of p38 MAPK at Threonine 180 (T180) and Tyrosine 182 (Y182). However, we noticed an increased phosphorylation at T222 of its substrate, MK2 (*Figure 2i*). The latter result could be due to an activation of the ERK pathway that might attenuate EZRIN/EGRF-dependent reduction of p38 MAPK signaling. Indeed, ERK2 binds and phosphorylates MK2 (*Sok et al., 2020*). Future studies will be needed to investigate ERK1/2 signaling is part of the EZRIN/EGFR-mediated signaling network. Taken together, these data suggest a model by which the EZRIN interaction with EGFR contributes to EGFR trafficking and signaling.

## Ezrin regulates endocytic EGFR sorting and signaling

Activation of EGFR leads to its internalization and trafficking to early endosomes, which sustains specific EGFR signaling and recycling (*Burke et al., 2001*). We asked if the increased EGFR protein level at cell membrane and the reduction of EGFR signaling in EZR$^{-/-}$ cells could be due to an alteration in EGFR dimerization and packaging into endosomal vesicles. To test this hypothesis, the cellular internalization and trafficking of EGFR basis was investigated by immunofluorescence and live-imaging studies. We found that the lack of Ezrin statistically reduced dimerization of EGFR upon EGF stimulation (*Figure 3a, b*). These results were confirmed by immunofluorescence analyses; indeed, compared to control cells, Ezrin-depleted cells showed higher levels of EGFR on the cell surface, which was mirrored by reduced EGFR abundance at endosomal compartments, as assessed by a reduction in the overlap between EGFR and EEA1 signals (*Figure 3c, d*) and increased EGFR protein levels on purified membranes (*Figure 3e, f*, upper panels). Consistently, EGFR endosomal localization was present on purified endosomes from control but not from EZR$^{-/-}$ cells (*Figure 3e, f*, lower panels). The reduced internalization of EGFR to endosomes was not accompanied by a suppression of endocytosis, as indicated by the slight and significant increase in the number of EEA1-positive early endosomes and endotracker-positive structures in EZR$^{-/-}$ compared to WT cells (*Figure 3g, h*). These results support that EGFR accumulation at the plasma membrane was not a result of an endocytosis defect. To better define whether the lack of EZRIN alters EGFR internalization and trafficking in EZR$^{-/-}$ cells upon EGF

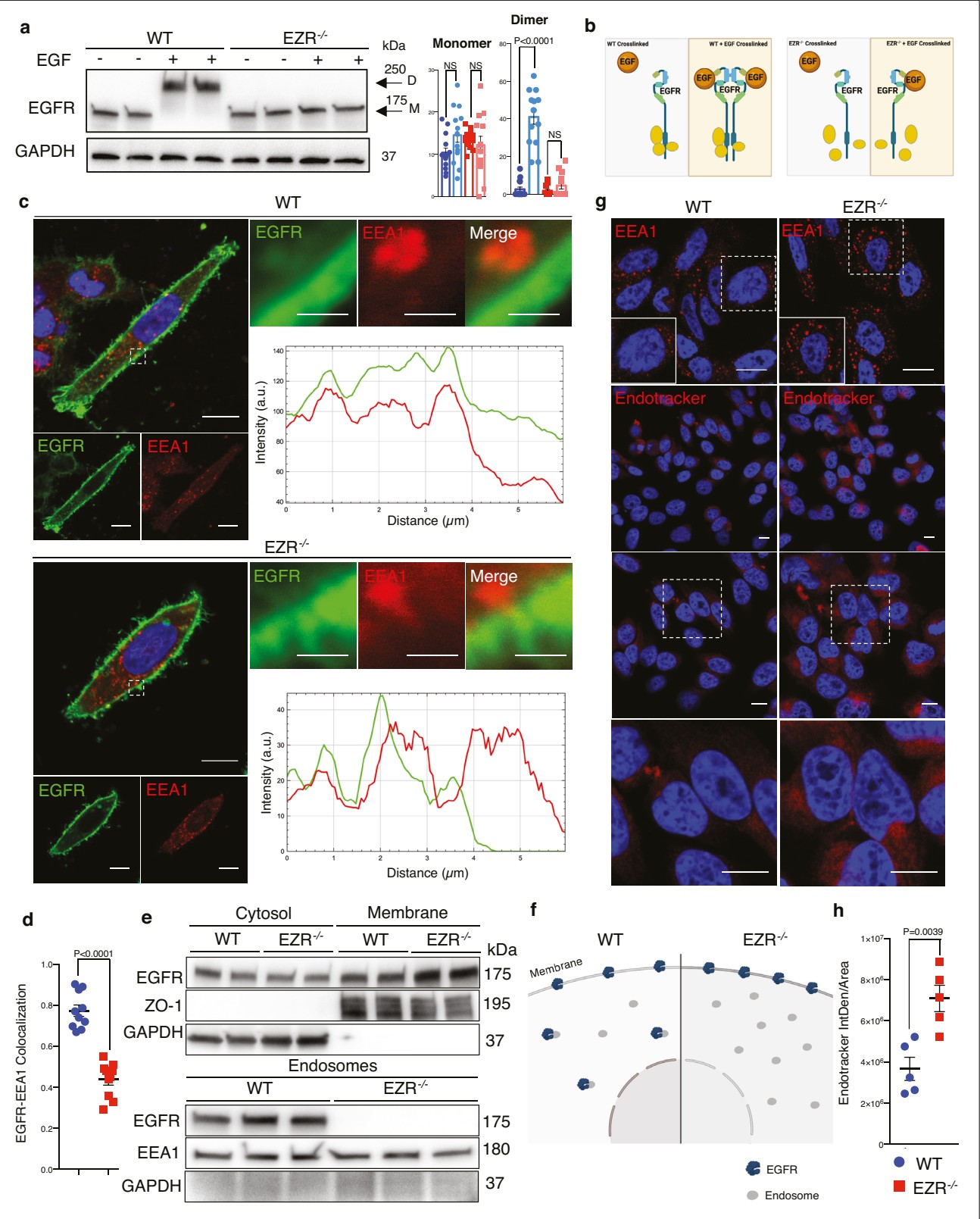

**Figure 3.** Ezrin controls EGFR localization. (**a**) Western blot analysis of chemical crosslinked EGFR in HeLa WT and EZR⁻/⁻ with (+) and without (−) EGF stimulation. Arrowheads indicate detected signals of dimeric and monomeric form of EGFR. (**b**) Model showing the crosslinking effect of EGFR dimer formation in HeLa WT and EZR⁻/⁻. This panel was created using BioRender.com. (**c**) Immunofluorescence images of EGFR (green) and EEA1 (red) in HeLa WT (top) and EZR⁻/⁻ (bottom) observed by confocal microscopy. Scale bar: 10 µm (magnification 1 µm). Representative plots of co-localization profiles of

*Figure 3 continued on next page*

*Figure 3 continued*

EGFR in early endosome. HeLa EZR$^{-/-}$ cells do not show EGFR and EEA1 co-localization compared to control cells. (**d**) Data represent mean of EGFR–EEA1 co-localization spots ± SEM (*n* = 3 experiments at least). Statistical test: unpaired *t*-test. (**e**) Representative immunoblots of EGFR in membrane (top) and endosomes (bottom) proteins in HeLa WT and EZR$^{-/-}$. ZO-1 and EEA1 are used as membrane and endosomes extraction control, respectively. GAPDH is used as loading control. (**f**) Schematic translocation of EGFR in the endosomes in HeLa WT compared to HeLa EZR$^{-/-}$. This panel was created using BioRender.com. (**g**) HeLa cells were fixed and immunostained with endotracker and EEA1 (red) and DAPI (blue). Scale bar: 10 µm (magnification 1 µm). (**h**) Graph shows mean of endotracker-positive cells ± SEM (*n* = 3 experiments at least). Statistical test: unpaired *t*-test.

The online version of this article includes the following source data for figure 3:

**Source data 1.** Raw uncropped and unedited blots relating to *Figure 3*.

**Source data 2.** Uncropped blots with the relevant bands labeled relating to *Figure 3*.

stimulation, we performed Total Internal Reflection Fluorescence (TIRF) time-lapse imaging at high spatiotemporal resolution. Both EZR$^{-/-}$ and control cells, transfected with an EGFR-GFP vector, were imaged every 0.5 s for 5 min upon EGF treatment. Notably, the EGF-induced EGFR endosomal internalization was dramatically abolished in EZR$^{-/-}$ compared to control cells (*Figures 4a-c and 5a*, and *Figure 5—videos 1–4*). Consistent with a defective EGFR integration in the early endosome, EGFR was localized at the plasma membrane in EZR$^{-/-}$ cells, despite EGF stimulation (*Figure 4a–c* and *Figure 5—videos 1–4*). These results suggest that EZRIN play an important role for the dimerization, integration, and trafficking of EGFR in the endosomes. To strengthen these findings, we performed ultrastructural analysis using immunoelectron microscopy (IEM) that further revealed the reduced number of EGFR-positive endosomal compartments and the increased presence of EGFR at the plasma membrane in EGF-stimulated EZR$^{-/-}$ compared with EGF-stimulated control cells (*Figure 5b*). Consistent with this, pY845 EGFR, pY1068 EGFR, pT202/Y204 p44/42 MAPK, and pT180/pY182 p38 MAPK were reduced upon EGF stimulation in EZR$^{-/-}$ cells (*Figure 5c*). Moreover, the increased EGFR internalization from membranes to endosomes by EGF stimulation was significantly inhibited in EZR$^{-/-}$ cells compared to WT (*Figure 5d*). As expected, we found that the endosomal EGFR internalization was further repressed in MEF-EZR$^{KO}$ (*Figure 2—figure supplement 1d–h*) and in HeLa cells upon NSC668394 treatment (*Figure 5—figure supplement 1a, b*), a specific Ezrin inhibitor (*Naso et al., 2020*). Taken together, these data strongly support a primary role of EZRIN in mediating the internalization and trafficking of EGFR from plasma membrane to endosomes.

## The endosomal Ezrin–EGFR complex targets TSC complex protein

We next sought to identify the molecular networks by which Ezrin/EGFR axis controls lysosomal biogenesis and function. Interestingly, EGFR stimulates several downstream effectors, including PI3K/AKT signaling in response to multiple stimuli (*Wee and Wang, 2017*). This led us to investigate the role of Ezrin/EGFR axis in the control of AKT signaling. AKT binds, phosphorylates, and inhibits hamartin (TSC1) and tuberin (TSC2) complex. TSC complex is essential to turn off the activity of Rheb, a crucial activator of mTORC1 at lysosomal surface (*Dibble and Cantley, 2015*). This raised the possibility of Ezrin-mediated activation of EGFR signaling would be required for AKT activation and thus stimulation of the mTORC1 pathway via TSC complex repression. To test this hypothesis, we analyzed the interaction between endogenous Ezrin/EGFR with AKT and TSC1 to define an endosomal signaling platform. In agreement with previously presented data (*Haddad et al., 2002*), TSC1and AKT co-immunoprecipitated with EZRIN (*Figure 6a*, upper panel). We also noticed that Ezrin was able to interact with TSC2 (*Figure 6a*, upper panel). The molecular basis of these interactions was investigated by in silico domain–domain interaction analyses. Accordingly, EZRIN (UID:P15311) was found as a scaffold protein interacting through its FERM central domain with the PK domain of EGFR, as detailed above (*Figure 2—figure supplement 1c*), and binding TSC1. Although Alphafold3 modeling of the EZRIN/TSC1 dimer did not provide high-confidence results, suggesting that the TSC1 (PF04388) could interact with both FERM N-terminal (PF09380) and C-terminal (PF09379) domains of EZRIN (*Figure 5—figure supplement 1c–h*). Concordantly, PPIDomainMiner identifies the FERM-N/C-hamartin as moderately confident (silver class), further supporting the possibility of EGFR/EZRIN/TSC1 interactions. Consistently, immunoprecipitation of EGFR was able to pull down both TSC1 and AKT (*Figure 6a*, lower panel), suggesting that EGFR, AKT, TSC1, and EZRIN are present in a complex. The latter results led us to investigate whether EGFR could interact with AKT and TSC1 indirectly through EZRIN. Co-immunoprecipitation experiments confirmed this possibility, given that *EZRIN* depletion abolished the

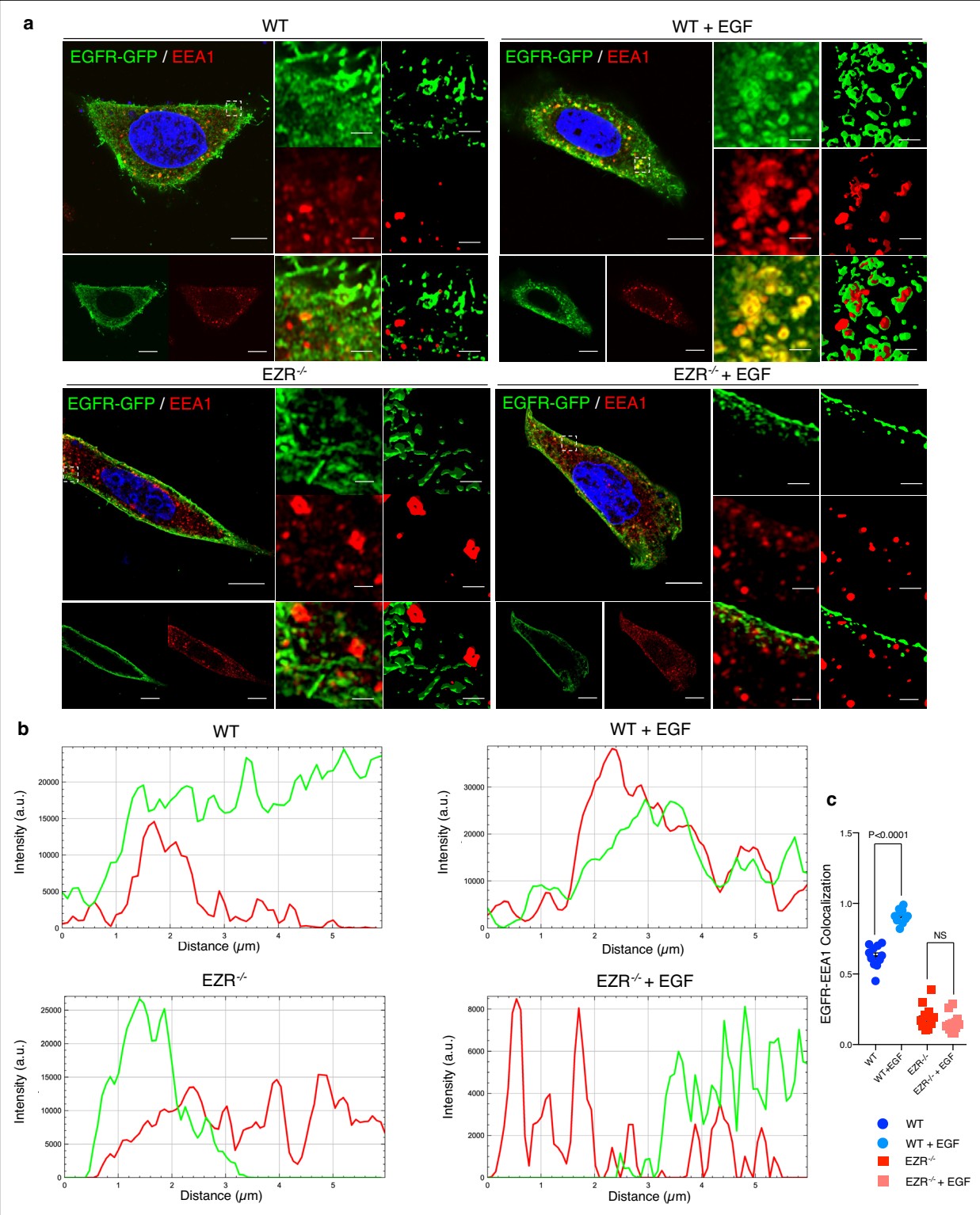

**Figure 4.** EGFR migrates on the endosomes depending on Ezrin. (**a**) Immunofluorescence labeling images of EGFR-GFP (green), EEA1 (red), and DAPI (blue) after 3 hr of EGF stimulation (right) in HeLa WT (top) and EZR$^{-/-}$ (bottom). Magnified views of the regions in the boxes are provided in both Airyscan high-resolution microscopy and 3D-confocal microscopy. (**b**) EGFR and EEA1 co-localization is expressed as a representative plot in HeLa WT (top) and EZR$^{-/-}$ (bottom). Scale bar: 10 μm (magnification 1 μm). (**c**) Data represent mean of EGFR–EEA1 co-localization spots ± SEM (*n* = 3 experiments at least). Statistical test: one-way ANOVA.

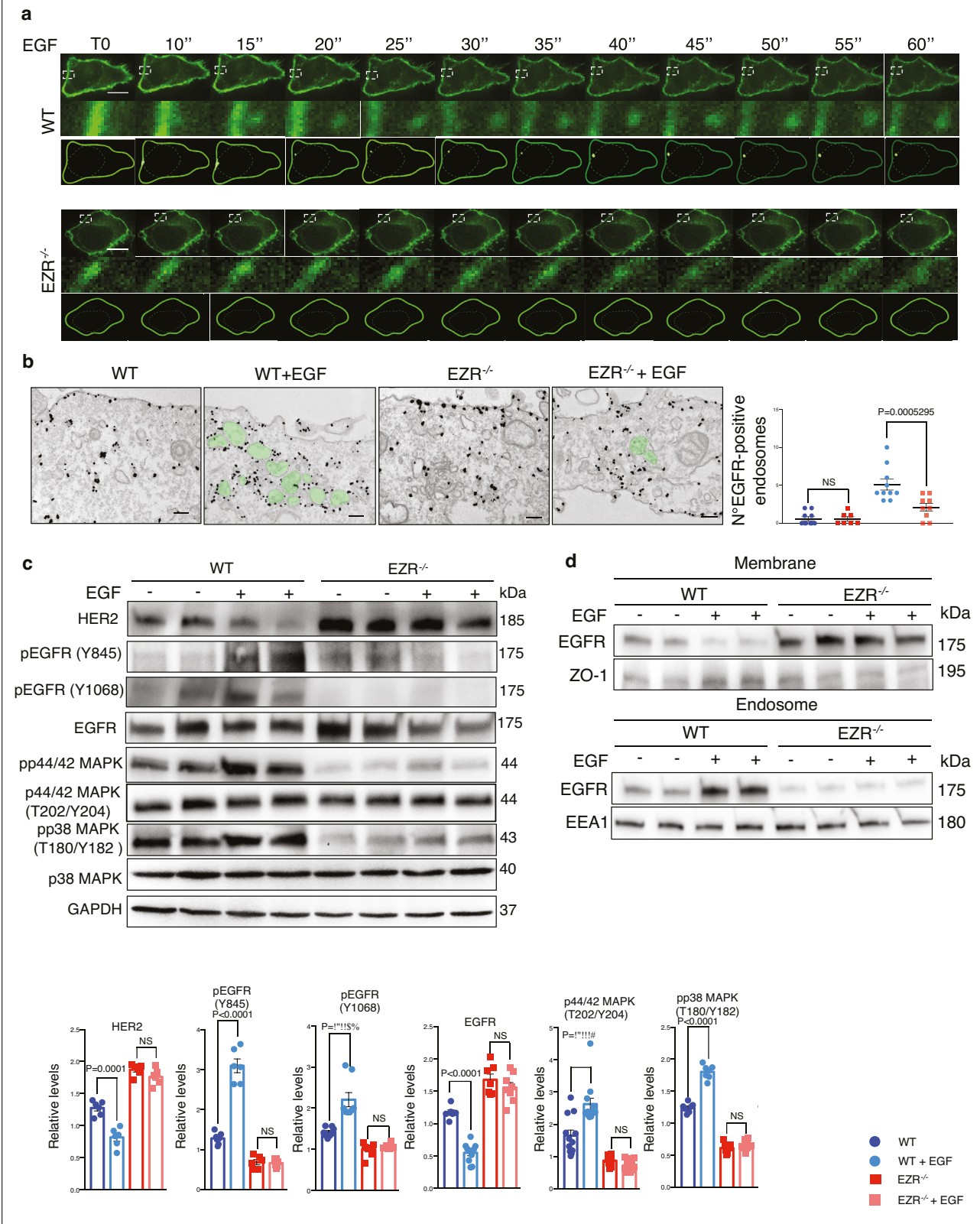

**Figure 5.** EGF stimulation does not affect EGFR in absence of Ezrin. (**a**) Live cell imaging and model for EGFR (green) translocation from the membrane to the endosomes in HeLa WT (top) and EZR$^{-/-}$ (bottom) cells without EGF stimulation (T0) and with a progressive EGF stimulation (from T10" to T60"). White boxes are magnifications that depict EGFR protein migration. Scale bar: 1 μm. Please refer to *Figure 5—video 1*. (**b**) IEM (anti-GFP immunolabeling) of cycloheximide-treated HeLa WT, WT + EGF, EZR$^{-/-}$, and EZR$^{-/-}$ + EGF cells expressing EGFR-GFP. Endosomes containing EGFR

*Figure 5 continued on next page*

*Figure 5 continued*

is shown in green. Scale bar: 200 nm. Quantitative analysis (right) of EGFR-positive endosomes expressed as mean ± SEM. Statistical test: generalized linear model with likelihood ratio (Poisson regression). (**c**) Immunoblots and calculated levels (bottom) of HER2, pY845 EGFR, pY1068 EGFR, EGFR, pT202/Y204 p42/44 MAPK, pT180/pY182 p38 MAPK, and P38 MAPK in HeLa WT and EZR$^{-/-}$ cells with (+) and without (−) EGF stimulation. Data are expressed as mean of pY845EGFR/EGFR and pT180/pY182 p38 MAPK/P38 MAPK ratio ± SEM (*n* = 3 experiments at least). GAPDH was used as loading control. Statistical test: unpaired *t*-test for HER2 WT, HER2 EZR$^{-/-}$, pY845 EGFR EZR$^{-/-}$, pY1068 EGFR WT, pY1068 EGFR EZR$^{-/-}$, EGFR WT, EGFR EZR$^{-/-}$, pT202/Y204 p44/42 MAPK WT, pT202/Y204 p44/42 MAPK EZR$^{-/-}$, pT180/pY182 p38 MAPK WT, and pT180/pY182 p38 MAPK EZR$^{-/-}$; unpaired *t*-test with Welch's correction for pY845 EGFR WT. (**d**) Representative immunoblots of EGFR in membrane (top) and endosomes (bottom) proteins in HeLa WT and EZR$^{-/-}$ with (+) and without (−) EGF stimulation. ZO-1 and EEA1 are used as membrane and endosomes extraction control, respectively. GAPDH is used as loading control.

The online version of this article includes the following video, source data, and figure supplement(s) for figure 5:

**Source data 1.** Raw uncropped and unedited blots relating to *Figure 5*.

**Source data 2.** Uncropped blots with the relevant bands labeled relating to *Figure 5*.

**Figure supplement 1.** EZRIN interacts with TSC1.

**Figure supplement 1—source data 1.** Raw uncropped and unedited blots relating to *Figure 5—figure supplement 1*.

**Figure supplement 1—source data 2.** Uncropped blots with the relevant bands labeled relating to *Figure 5—figure supplement 1*.

**Figure 5—video 1.** WT HeLa cells expressing EGFR-GFP were imaged by Total Internal Reflection Fluorescence (TIRF) super-resolution microscopy every were imaged every 0.5 s for 5 min after EGF stimulation (related to *Figure 5a*).

https://elifesciences.org/articles/98523/figures#fig5video1

**Figure 5—video 2.** Magnification from WT HeLa cells expressing EGFR-GFP, imaged by Total Internal Reflection Fluorescence (TIRF) super-resolution microscopy every were imaged every 0.5 s for 5 min after EGF stimulation (related to *Figure 5a*).

https://elifesciences.org/articles/98523/figures#fig5video2

**Figure 5—video 3.** EZR$^{-/-}$ HeLa cells expressing EGFR-GFP were imaged by Total Internal Reflection Fluorescence (TIRF) super-resolution microscopy every were imaged every 0.5 s for 5 min after EGF stimulation (related to *Figure 5a*).

https://elifesciences.org/articles/98523/figures#fig5video3

**Figure 5—video 4.** Magnification from EZR$^{-/-}$ HeLa cells expressing EGFR-GFP, imaged by Total Internal Reflection Fluorescence (TIRF) super-resolution microscopy every were imaged every 0.5 s for 5 min after EGF stimulation (related to *Figure 5a*).

https://elifesciences.org/articles/98523/figures#fig5video4

interactions of EGFR with TSC1 and AKT, thereby pointing out the role of Ezrin as a scaffold protein for the formation and activation of the EGFR/AKT/TSC1 signaling (*Figure 6a*). In agreement with this hypothesis, the lack of Ezrin reduced pS473 AKT activation and in turn suppressed AKT-mediated phosphorylation of pS939 TSC2 (*Figure 6b*). As expected, the inactivation of AKT promoted activation of TSC1 and TSC2, which localized on the lysosomes in EZR$^{-/-}$ cells (*Figure 6d, e*). Consistently, translocation of the TSC complex on the lysosomes led to inhibition of mTORC1 pathway, as demonstrated by reduction of pT389 P70 S6 Kinase and pS65 4E-BP1 levels (*Figure 6b*). In agreement, the insulin administration was not able to restore mTORC1 signals (*Figure 6c*) and lysosomal localization of TSC complex in EZR$^{-/-}$ cells (*Figure 5—figure supplement 1j*), in line with the hypothesis that Ezrin acts as a scaffold protein for AKT/TSC complex. Thus, we ensured that these findings were also confirmed when EZRIN was pharmacologically inhibited on HeLa (*Figure 6—figure supplement 1a*) and ARPE-19 cells (*Figure 6—figure supplement 1b, c*). Concordantly, these results were mirrored in MEF EZR$^{KO}$ cells that yield a similar pattern (*Figure 6—figure supplement 1d, e*). To strengthen the molecular mechanism by which EGFR/EZRIN controls mTORC1 pathway via TSC complex, we investigated the effects of EZRIN inhibition on MEF TSC2$^{KO}$ cells and found that depletion of TSC2 rescue TORC1 signaling when Ezrin was pharmacologically inhibited (*Figure 6—figure supplement 1f*). Together, these results establish an Ezrin-dependent molecular machinery coordinating EGFR sorting and signaling at the endosome to a well-regulated signals transfer to lysosomes via AKT/TSC complex axis. Consistent with this idea, phosphorylation of pS473 AKT was significantly abolished in response to EGF treatment in EZR$^{-/-}$ compared to control cells (*Figure 6f*). Accordingly, using confocal Airyscan high-resolution microscopy, we found that the majority of the TSC complex was present in early endosomes of HeLa WT cells upon EGF treatment, as shown by co-localization with the endosomal marker EEA1 (*Figure 6g, h*). Notably, the endosomal TSC complex localization was abolished in EGF-treated EZR$^{-/-}$ cells (*Figure 6g, h*), which indicates that EGFR-mediated repression of TSC complex by AKT activation could occur in a stable endosomal complex dependent on Ezrin. Consistently, the ectopic

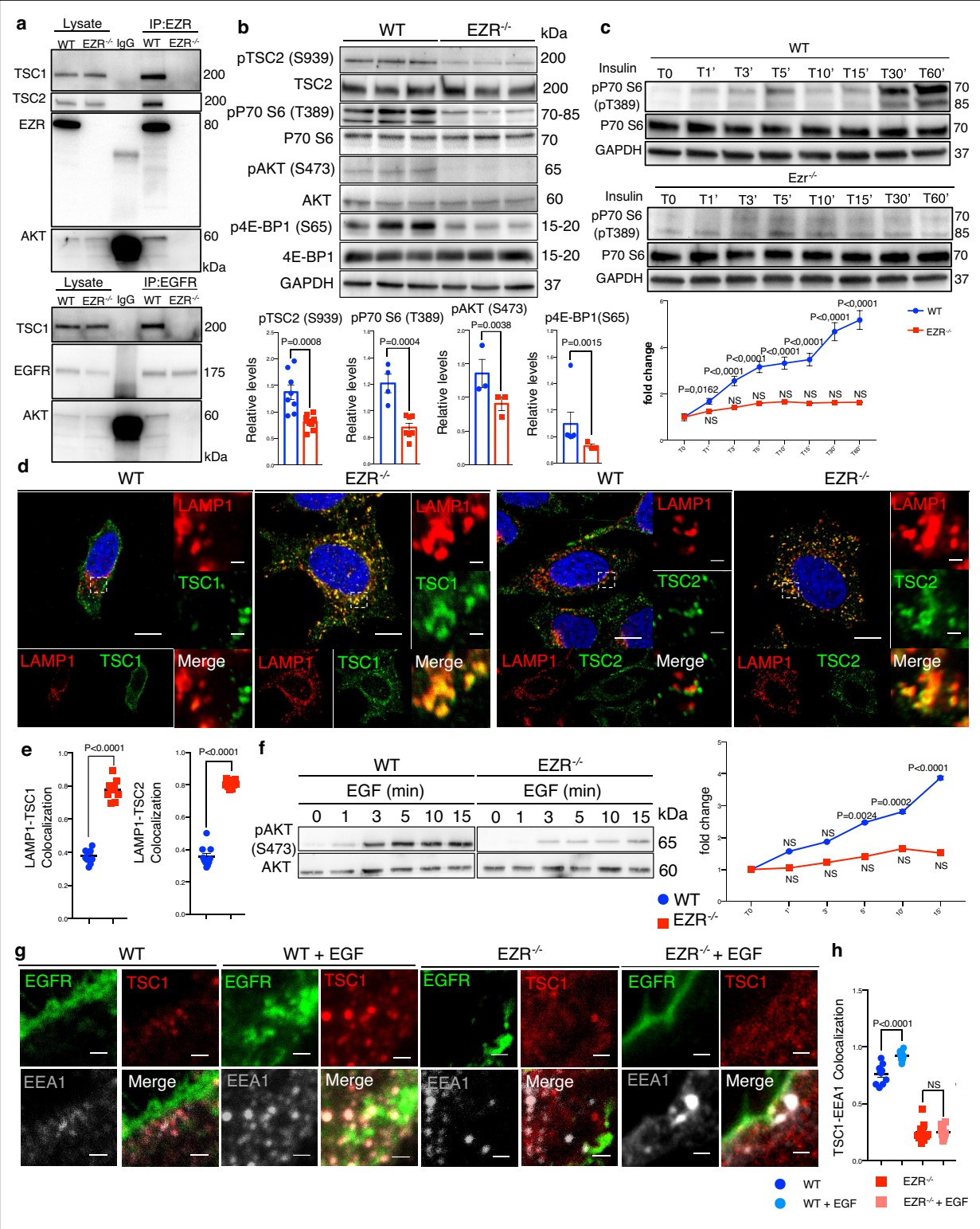

**Figure 6.** EGFR–Ezrin complex interacts with TSC1. (**a**) Co-IP analysis for Ezr–TSC1 (left) and EGFR–TSC1 (right) interaction. For co-IP analyses, Ezrin (left) and EGFR (right) antibodies were used. The proteins immunoprecipitated were blotted for TSC1 and AKT antibodies in HeLa WT and EZR⁻/⁻. (**b**) HeLa WT and EZR⁻/⁻ cells were lysed and immunoblotted with pS939 TSC2, TSC2, PT389 P70 S6 Kinase, P70 S6 Kinase, pS473 AKT, AKT, pS65 4E-BP1, 4E-BP1, and GAPDH as a loading control. Data represent the mean of pS939 TSC2/TSC2, T389 P70 S6 Kinase/P70 S6 Kinase, pS473 AKT/AKT, and pS65 4E-BP1/4E-BP1 ratio ± SEM (*n* = 3 experiments at least). Statistical test: unpaired *t*-test for pT389 P70 S6 Kinase, pS473 AKT; unpaired *t*-test with Welch's correction for pS939 TSC2; Mann–Whitney test for pS939 TSC2. (**c**) pP70 S6 Kinase western blotting with insulin time course in HeLa WT

*Figure 6 continued on next page*

*Figure 6 continued*

(up) and EZR$^{-/-}$ (bottom) cells. Graph shows the mean of pP70 S6/P70 S6 ratio ± SEM (*n* = 3 experiments at least). Statistical test: one-way ANOVA for WT and KO curve (pairwise comparisons with reference T0). (**d**) Representative confocal images of LAMP1 and TSC1 (left) and LAMP1 and TSC2 (right) immunofluorescence in HeLa WT and EZR$^{-/-}$ cells. Magnified insets of TSC1/2 localization are shown. Scale bar: 10 μm (magnification 1 μm). (**e**) Data represent mean of LAMP1–TSC1 (left) and LAMP1–TSC2 (right) co-localization spots ± SEM (*n* = 3 experiments at least). Statistical test: unpaired *t*-test for LAMP1–TSC1; unpaired *t*-test with Welch's correction for LAMP1–TSC2. (**f**) pS473 AKT western blotting with EGF time course in HeLa WT (left) and EZR$^{-/-}$ (right) cells. Graph shows the mean of pS473 AKT/AKT ratio ± SEM (*n* = 3 experiments at least). Statistical test: one-way ANOVA with Dunnett's post hoc test for WT curve; Kruskal–Wallis test with Dunn's post hoc test for KO curve (pairwise comparisons with reference T0). NS: not significant. (**g**) HeLa WT, WT + EGF, EZR$^{-/-}$, and EZR$^{-/-}$ + EGF cells were immunostained with EGFR (green), TSC1 (red), and EEA1 (gray). Representative magnifications are shown. Scale bar: 10 μm (magnification 1 μm). (**h**) Data represent mean of TSC1–EEA1 co-localization spots ± SEM (*n* = 3 experiments at least). Statistical test: one-way ANOVA.

The online version of this article includes the following source data and figure supplement(s) for figure 6:

**Source data 1.** Raw uncropped and unedited blots relating to *Figure 6*.

**Source data 2.** Uncropped blots with the relevant bands labeled relating to *Figure 6*.

**Figure supplement 1.** Ezrin inhibition induces mTORC1C1 pathway inhibition.

**Figure supplement 1—source data 1.** Raw uncropped and unedited blots relating to *Figure 6—figure supplement 1*.

**Figure supplement 1—source data 2.** Uncropped blots with the relevant bands labeled relating to *Figure 6—figure supplement 1*.

**Figure supplement 2.** Ezrin overexpression rescue EGFR and TSC1 localization.

expression of a constitutively active EZR$^{T567D}$ protein, but not a constitutively inactive Ezr$^{T567A}$ protein, rescued EGFR sorting and signaling activation at the endosome in EZR$^{-/-}$ cells upon EGF treatment (*Figure 6—figure supplement 2a*). Additionally, EZR$^{T567D}$, but not EZR$^{T567A}$, rescued the physiological localization of TSC complex on the cytoplasm in EZR$^{-/-}$ cells (*Figure 6—figure supplement 2b*). Moreover, full translocation of TSC complex on endosomes was restored in EZR$^{-/-}$ cells expressing EZR$^{T567D}$ protein after EGF treatment, as shown by co-localization between TSC1 and EEA1 proteins (*Figure 6—figure supplement 2b*).

## Aberrant EGFR signaling induces retinal degeneration in EZR$^{-/-}$ medaka fish

To further investigate the role of EZRIN/EGFR axis, which is conserved among vertebrates, in daily modulation of lysosomal biogenesis and function in retinal cells, we carried out in vivo experiments. Accordingly, we found that the EGFR expression pattern in the rodents' retina diminished in response to light and increased after light off (*Figure 7—figure supplement 1a, b*), coinciding with Ezrin expression and diurnal lysosomal biogenesis in the RPE/retina (*Naso et al., 2020*). Consistently, we found an inhibition of TSC2 and an increase of AKT/mTORC1 pathway in the mice retina in response to dark condition, when active Ezrin (*Naso et al., 2020*) and EGFR are highly expressed (*Figure 7—figure supplement 1c*). Moreover, we found that TSC2 was dephosphorylated in response to light in the retina, when inactive Ezrin (*Naso et al., 2020*) and EGFR are weakly expressed (*Figure 7—figure supplement 1c*) as a consequence of a decrease of the AKT/mTORC1 signaling, which suggests that activation of Ezrin underlies the requirement endosomal EGFR signaling to assemble the EGFR/AKT/TSC complex and represses lysosomal biogenesis. This data supported that EGFR signaling in retinal cells could be regulated by Ezrin for finely control lysosomal biogenesis and function in mTORC1-dependent manner. Thus, we used the highly effective CRISPR–Cas9-mediated mutagenesis to create stable Ezrin mutant lines in Medaka fish (*Oryzias latipes*, Ol) as in vivo model system. Targeting two sgRNAs (sgRNA1 and sgRNA2) in the exon 1 of *Ezrin* gene, we generated a 386-bp deletion and established founder lines for this deletion. This mutation eliminates the first 129 amino acids containing ATG (olEzrin$^{Δ386}$), generating a severely truncated Ezrin protein (*Figure 7a*), which was not detectable by western blot analyses (*Figure 7b*). This indicated that the Δ386 Ezrin allele is likely functionally null, and mutants will hereafter be called Ezrin$^{-/-}$ medaka line. Larval homozygous Ezrin$^{-/-}$ medaka line appeared almost visually indistinguishable from wild-type siblings, and the Δ386 allele was inherited in Mendelian ratios (*Figure 7c*). Interestingly, the Ezrin$^{-/-}$ medaka larvae recapitulated in part a previously reported phenotype characterized in postnatal ezrin knockout mice (*Bonilha et al., 2006*). Consistent with our in vitro data, we observed increased levels of the EGFR protein accompanied by a significant reduction of active pY845 EGFR. This was associated by a significant

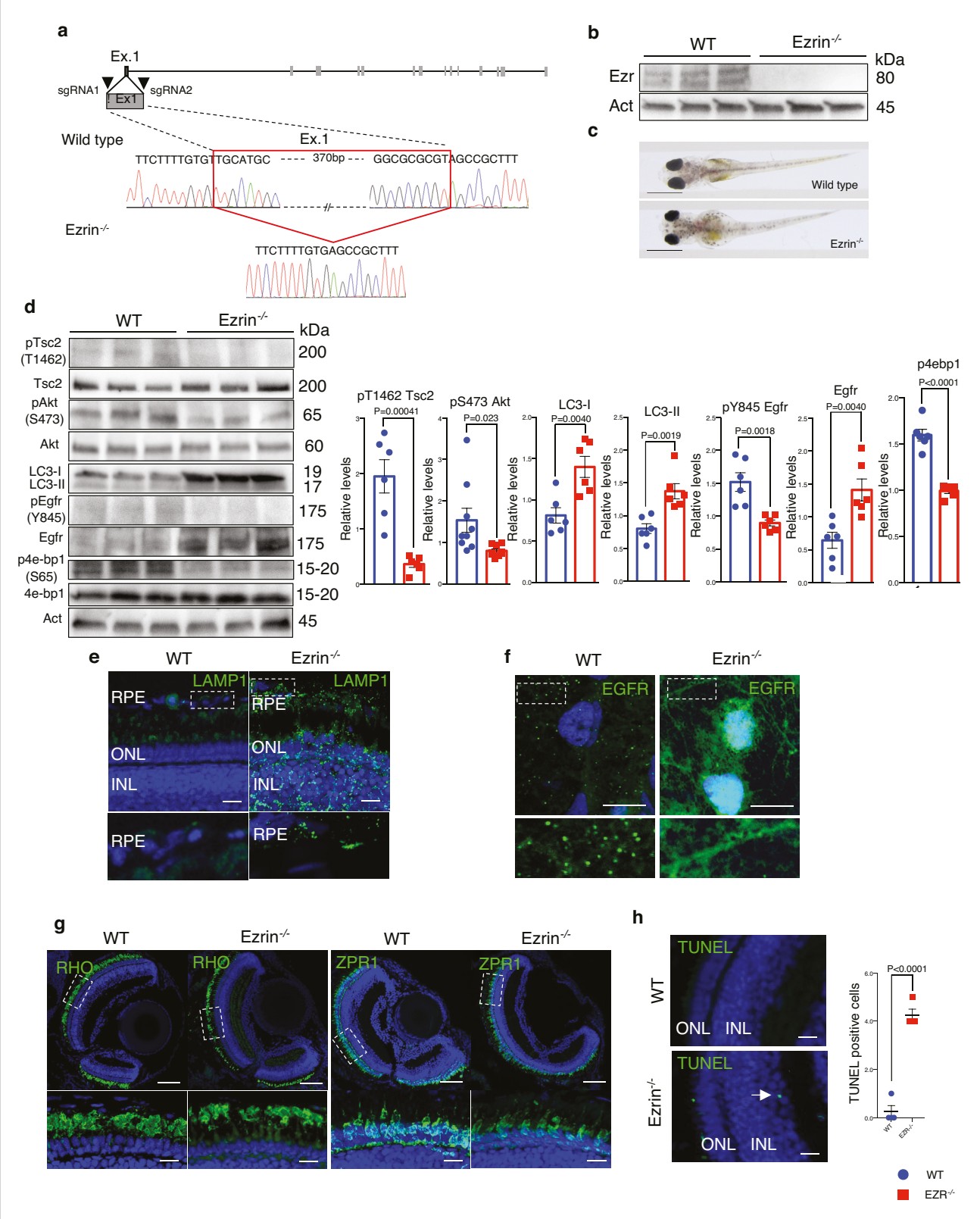

**Figure 7.** *Ezrin* depletion induces EGFR-mediated retinal degeneration. (**a**) Schematic representation of used CRISPR/Cas9 strategy to generate Ezrin⁻/⁻ medaka lines. The red box highlighted the deleted nucleotides in the *Ezrin* exon 1 gene. (**b**) WT and Ezrin⁻/⁻ medaka proteins were immunoblotted with Ezrin antibody and Actin as a loading control. (**c**) Stereo-microscopic representative images of WT and Ezrin⁻/⁻ medaka at stage 40. Scale bar: 1 mm. (**d**) Immunoblots and calculated levels (right) of pT1462 TSC2, pS473 Akt, LC3-I, LC3-II, pY845 Egfr and Egfr, pS65 4E-BP1 in WT and Ezrin⁻/⁻ medaka

*Figure 7 continued*

fish. Data are expressed as mean of pT1462 TSC2/TSC2, pS473 Akt/Akt, pS65 4E-BP1/4EB-P1, and pY845EGFR/EGFR ratio ± SEM (*n* = 3 experiments at least). Actin was used as loading control. Statistical test: unpaired *t*-test. (**e**) Representative confocal images of LAMP1 immunofluorescence in WT and Ezrin⁻/⁻ medaka fish. Magnified insets of RPE LAMP1 localization are shown. Scale bar: 10 μm. RPE: retinal pigment epithelium; ONL: outer nuclear layer; INL: inner nuclear layer. (**f**) Medaka WT and Ezrin⁻/⁻ fish were immunostained with EGFR. Scale bar: 10 μm. (**g**) Immunofluorescence labeling images of RHO (left) and ZPR1 (right) in WT and Ezrin⁻/⁻ fish. Magnified views of the regions in the boxes are provided at the bottom. Scale bar: 10 μm. (**h**) Confocal images showing representative TUNEL-positive cells on cryosection from WT and Ezrin⁻/⁻ medaka lines. Scale bar: 10 μm. Graph shows the mean of number of TUNEL-positive cells for retina ± SEM (*n* = 3 experiments at least). Statistical test: unpaired *t*-test.

The online version of this article includes the following source data and figure supplement(s) for figure 7:

**Source data 1.** Raw uncropped and unedited blots relating to *Figure 7*.

**Source data 2.** Uncropped blots with the relevant bands labeled relating to *Figure 7*.

**Figure supplement 1.** Light/dark transitions regulate EGFR and mTORC11 signaling in mice retinal pigment epithelium (RPE).

**Figure supplement 1—source data 1.** Raw uncropped and unedited blots relating to *Figure 7—figure supplement 1*.

**Figure supplement 1—source data 2.** Uncropped blots with the relevant bands labeled relating to *Figure 7—figure supplement 1*.

reduction of the phosphorylation of pS473 AKT in Ezrin⁻/⁻ Medaka line compared to control larvae (*Figure 7d*). Consistently, we observed reduced AKT-mediated phosphorylation of pT1462 of TSC2, decreased mTORC1 signaling as shown by reduction of p4EBP1 (S65) and an increased autophagy, as demonstrated by higher levels of LC3-II and Lamp1 (*Figure 7d, e*). Notably, endosomal internalization of EGFR was significantly repressed in the RPE of Ezrin⁻/⁻ medaka line. Consistent with defective EGFR internalization and trafficking, whole-mount immunofluorescence analysis showed that EGFR accumulated at plasma membrane of RPE of Ezrin⁻/⁻ medaka line compared with control fish (*Figure 7f*). Considering the role of endosomal sorting and signaling in the health of retinal cells (*Toops et al., 2014*), we addressed the consequences of aberrant EGFR signaling pathway in retina of Ezrin⁻/⁻ medaka line. Notably, defective endosomal EGFR signaling was sufficient to induce deleterious consequences for the health of photoreceptor cells, which showed reduction in POS compared with native rods (*Figure 7g*), similar to the pathogenesis of macular degeneration (*Borrelli et al., 2020*; *Kaur and Lakkaraju, 2018*). Notably, depletion of *olEzrin* was also associated with a significant increase in the number of TUNEL-positive cells in the retina from Ezrin⁻/⁻ medaka compared to control (*Figure 7h*). Altogether, these data support the dynamic regulation of EGFR signaling at endosomal compartments in response to Ezrin activation, which assembles and activates an EGFR/AKT/TSC complex signalosome at endosomes to finely regulate the lysosomal signaling by mTORC1 pathway, required for the correct autophagy and retinal cell health (*Figure 8*).

## Discussion

Canonical EGFR signaling begins at the plasma membrane with the engagement of the EGF ligand (*Tanaka et al., 2018*). Emerging studies have indicated that sorting of the EGF–EGFR complex to endosomal vesicles requires spatiotemporally defined encounters with distinct cytoskeleton platforms resulting in internalization, activation, maintenance, or termination of EGFR signaling (*Ceresa, 2012*; *Wang et al., 2002*). Consistent with this notion, it is now increasingly recognized that many molecules participating in signal transduction are central sorting hubs that coordinate signaling from and to different intracellular compartments, including early endosomes, late endosomes, phagosomes, and lysosomes (*Sorkin and von Zastrow, 2009*). However, molecular networks determining selective signal transduction from endosomes to lysosomes are not well defined. In this study, we demonstrated that EZRIN is a cytoskeleton scaffold protein aligned along internal membranes, and that this localization is essential for endosomal EGFR signal transduction to the TSC complex. Endosomal EGFR sorting and activation occurs mainly due to binding with EZRIN that facilitates dimerization and activation of the EGF–EGFR receptor complex, resulting in their recruitment to endosomes, followed by AKT activation that targets and inhibits the TSC complex. Indeed, time-lapse confocal imaging revealed that EGFR fails to be recruited to endosomal compartments upon EGF stimulation in the absence of Ezrin or in the presence of its inactive form (EZRIN^T567A). The loss of Ezrin function compromises EGFR-mediated AKT activation, which in turn reduces TSC complex inhibition resulting in TSC complex translocation to lysosomes where it constrains mTORC1 activity. Consistent with this, not

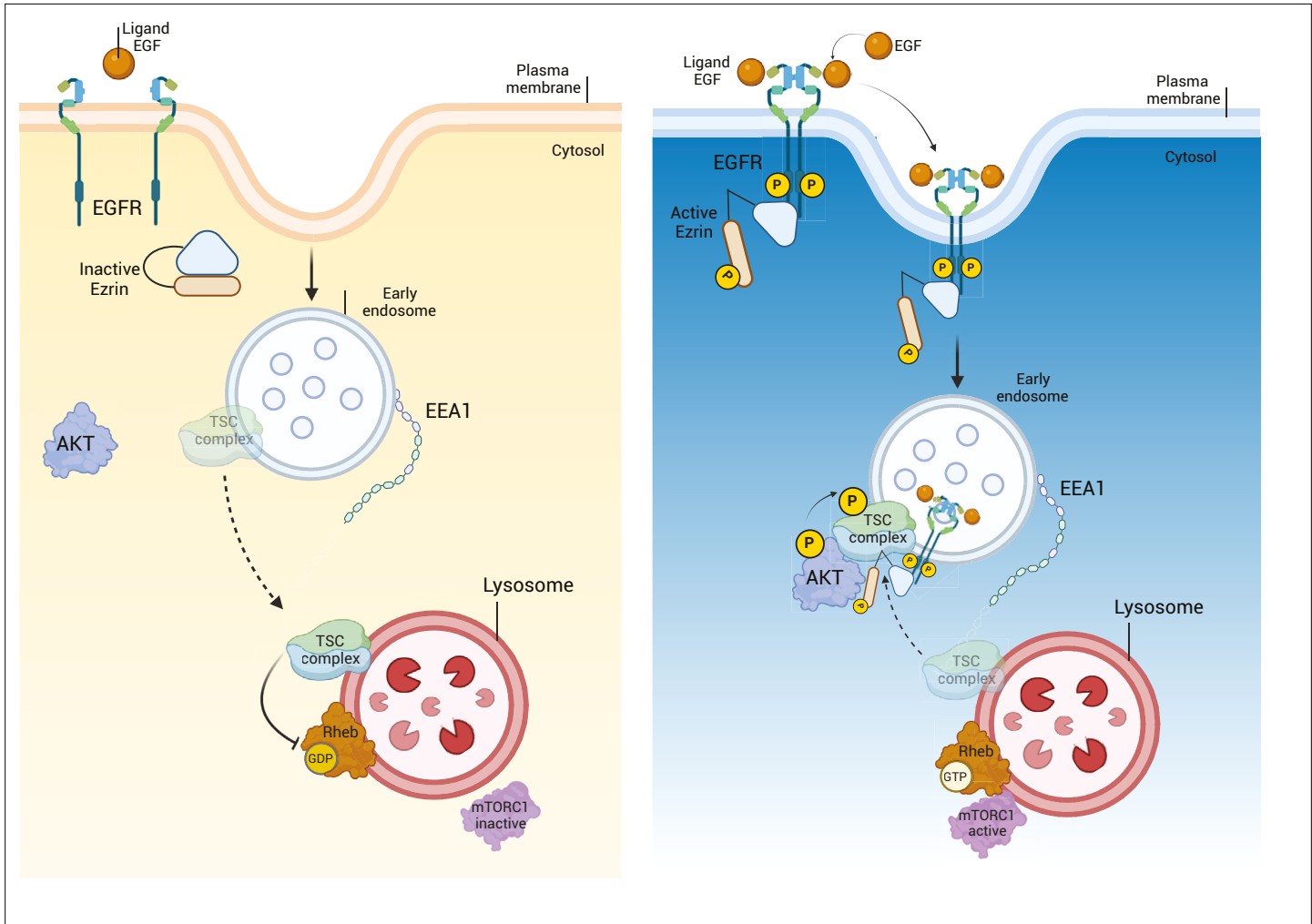

**Figure 8.** EGFR/Ezrin/TSC complex molecular pathway. Diurnal inactivation of Ezrin leads to incapacity of EGFR to dimerize. The absence of active EGFR on the endosome causes the migration of TSC complex on the lysosome, where it inhibits mTORC1C1. On the contrary, the nocturnal activation of Ezrin favorites the phosphorylation and dimerization of EGFR, that translocate from the plasma membrane on the early endosome. Ezrin localized with EGFR, on the endosomal membrane, binds TSC complex, preventing mTORC1C1 inactivation on the lysosome. This figure was created using BioRender.com.

only does loss of Ezrin impact on lysosomal TSC complex translocation, but we also documented that dephosphorylation of Ezrin in its inactive form is required for EGFR inactivation and TSC1 and TSC2 release from an EGFR/EZRIN complex, possibly to support the lysosomal biogenesis and function. Indeed, autophagy appears to be highly sensitive to pharmacological Ezrin inhibition via the EGFR/AKT axis. Moreover, overexpression of EZRIN[T567A], but not EZRIN[T567D], fails to restore EGFR endosomal signaling and lysosomal function in Ezrin-defective cells, indicating that phosphorylation of Ezrin is indispensable for this activity. Supporting this possibility, loss of βA3/A1-crystallin affects PITPβ/PLC signaling axis associated with an age-related loss of PLC-mediated Ezrin phosphorylation and subsequent compromised RPE cell polarity and EGFR signaling. Notably, the lysosome-mediated POS clearance was disrupted in the Cryba1 cKO RPE (*Shang et al., 2021*). Beyond this, our findings also showed an interaction of EGFR with the TSC complex and their co-localization with endosomes, opening to future work on mechanisms of how endosomal system connects extracellular signals with lysosomes under different physiological and pathological conditions.

Upregulation of Ezrin has been shown to induce an age-related macular degeneration-like phenotype in miR-211[−/−] mice (*Naso et al., 2020*), where light-mediated cell clearance is completely abolished. We can speculate that recruitment of EGFR on endosomal compartments by Ezrin orchestrates a local signal between endosomes and lysosomes to drive tight control on the lysosomal cargo

demands, although future studies are needed in this regard. Interestingly, inhibition of EGFR activity, by silencing Rubicon (RUBCN), switches lysosomal cargo degradation from POS- to LC3-associated phagocytosis to autophagy process in the RPE cells (*Muniz-Feliciano et al., 2017*). However, the molecular mechanisms are completely unknown. Therefore, Ezrin may represent a nodal point in endosomal compartments where EGFR signaling and AKT converge and integrate to directly control the TSC complex/mTORC1 pathway and lysosomal cargo demands and degradation. Notably, mutations affecting TSC1 and TSC2 alter lysosomal function with retinal manifestations in 40–50% of individuals (*Rosset et al., 2017*). This phenotype is also showed in RPE-specific deletion of TSC1 profoundly leading to an age-related impairment in lysosomal function associated with RPE degeneration in vivo (*Huang et al., 2019*).

The findings of the cellular mechanisms governing endosomal EGFR sorting and signaling might be of therapeutic relevance. Indeed, alteration of endosomal biogenesis and signaling have been shown to participate in the age-related, progressive neurodegeneration such as in age-related macular degeneration and Alzheimer's disease (*Kaur and Lakkaraju, 2018*). Thus, the identification of the mechanisms that control Ezrin/EGFR/mTORC1 molecular network might be exploited for the treatment of diseases in which defective endo-lysosomes play a part.

# Materials and methods

## Key resources table

| Reagent type (species) or resource | Designation | Source or reference | Identifiers | Additional information |
|---|---|---|---|---|
| Strain, strain background (*Oryzias latipes*, male and female) | *Oryzias latipes* | *Iwamatsu, 2004* | | |
| Strain, strain background (*Oryzias latipes*, male and female) | *Oryzias latipes* EZR$^{-/-}$ | This paper | | See Materials and methods: Ezrin$^{-/-}$ medaka generation by CRISPR/Cas9 system |
| Cell line (human) | ARPE-19 | ATCC | CRL-2302 | |
| Cell line (human) | HeLa | ATCC | CCL-2 | |
| Cell line (mouse) | MEF | ATCC | SCRC-1008 | |
| Cell line (human) | HeLa EZR$^{-/-}$ | This paper | | See Materials and methods: Generation of an EZR$^{-/-}$ HeLa cell line |
| Transfected construct (human) | Ezrin$^{T567D}$-mCherry | S.Coskoy lab (Institute Curie, Paris) | | |
| Transfected construct (human) | Ezrin$^{T567A}$-mCherry | S.Coskoy lab (Institute Curie, Paris) | | |
| Transfected construct (human) | EGFR-GFP | Addgene | 32751 | |
| Transfected construct (*Streptococcus pyogenes* M1) | pCS2-nCas9n | Addgene | #4729 | |
| Transfected construct | Lipofectamine 2000 | Invitrogen | 12566014 | |
| Antibody | anti-NBR1 (mouse monoclonal) | Abnova | MO1 | WB (1:1000) |
| Antibody | anti-LAMP1 (rat monoclonal) | Santa Cruz | Sc-19992 | IF (1:400) |
| Antibody | anti-LAMP1 (rabbit monoclonal) | Sigma | L1418 | WB (1:500) |
| Antibody | anti-LAMP1 (mouse monoclonal) | DSHB | H4A3 | IF (1:1000) |
| Antibody | anti-LAMP1 (rabbit polyclonal) | Abcam | ab24170 | IF (1:100) |
| Antibody | anti-Ezrin (mouse monoclonal) | Novex | 357300 | WB (1:1000) |
| Antibody | anti-SQSTM1/P62 (mouse monoclonal) | Abcam | ab56416 | WB (1:1000) |
| Antibody | anti-Cathepsin D (rabbit monoclonal) | Cell Signaling | 2284 | WB (1:1000) |

*Continued on next page*

*Continued*

| Reagent type (species) or resource | Designation | Source or reference | Identifiers | Additional information |
|---|---|---|---|---|
| Antibody | anti-LC3 (rabbit polyclonal) | Novus | NB100-2220 | WB (1:1000) IF (1:200) |
| Antibody | anti-GAPDH (mouse monoclonal) | Santa Cruz | SC-32233 | WB (1:1000) |
| Antibody | anti-HER2/ErbB2 (rabbit monoclonal) | Cell Signaling | 2165 | WB (1:1000) |
| Antibody | anti-HER3/ErbB3 (rabbit monoclonal) | Cell Signaling | 12708 | WB (1:1000) |
| Antibody | anti-EGF receptor (rabbit monoclonal) | Cell Signaling | 4267 | WB (1:1000) IF (1:50) |
| Antibody | anti-phospho-EGF receptor (Tyr1068) (rabbit monoclonal) | Cell Signaling | 3777 | WB (1:1000) |
| Antibody | anti-phospho-EGF receptor (Tyr845) (rabbit monoclonal) | Cell Signaling | 6963 | WB (1:1000) |
| Antibody | anti-MAPKAPK-2 (rabbit polyclonal) | Cell Signaling | 3042 | WB (1:1000) |
| Antibody | anti-phospho-MAPKAPK-2 (Thr222) (rabbit monoclonal) | Cell Signaling | 3316 | WB (1:1000) |
| Antibody | anti-p38 MAPK (rabbit monoclonal) | Cell Signaling | 8690 | WB (1:1000) |
| Antibody | anti-phospho-p38 MAPK (Thr180/Tyr182) (rabbit monoclonal) | Cell Signaling | 4511 | WB (1:1000) |
| Antibody | anti-ZO1 (rabbit polyclonal) | Abcam | ab216880 | WB (1:1000) |
| Antibody | anti-EEA1 (mouse monoclonal) | BD | 610457 | WB (1:1000) IF (1:100) |
| Antibody | anti-Tuberin/TSC2 (rabbit monoclonal) | Cell Signaling | 4308 | WB (1:1000) IF (1:100) |
| Antibody | anti-phospho-Tuberin/TSC2 (Ser939) (rabbit polyclonal) | Cell Signaling | 3615 | WB (1:1000) |
| Antibody | anti-phospho-Tuberin/TSC2 (Thr1462) (rabbit monoclonal) | Cell Signaling | 3617 | WB (1:1000) |
| Antibody | anti-p70 S6 Kinase (rabbit polyclonal) | Cell Signaling | 9202 | WB (1:1000) |
| Antibody | anti-phospho-p70 S6 Kinase (Thr389) (mouse monoclonal) | Cell Signaling | 9206 | WB (1:1000) |
| Antibody | anti-Akt (rabbit polyclonal) | Cell Signaling | 9272 | WB (1:1000) |
| Antibody | anti-phospho-Akt (Ser473) (rabbit monoclonal) | Cell Signaling | 4060 | WB (1:1000) |
| Antibody | anti-4E-BP1 (rabbit monoclonal) | Cell Signaling | 9644 | WB (1:1000) |
| Antibody | anti-phospho-4E-BP1 (Ser65) (rabbit monoclonal) | Cell Signaling | 9456 | WB (1:1000) |
| Antibody | anti-phospho-4E-BP1 (Thr37/46) (rabbit monoclonal) | Cell Signaling | 2855 | WB (1:1000) |
| Antibody | anti-Hamartin/TSC1 (rabbit monoclonal) | Cell Signaling | 6935 | WB (1:1000) IF (1:1000) |
| Antibody | anti-EGFR (mouse monoclonal) | Santa Cruz | sc-120 | WB (1:500) IF (1:50) |
| Antibody | anti-p-EGFR (mouse monoclonal) | Santa Cruz | sc-57542 | WB (1:500) |

*Continued on next page*

*Continued*

| Reagent type (species) or resource | Designation | Source or reference | Identifiers | Additional information |
|---|---|---|---|---|
| Antibody | anti-GFP (chicken monoclonal) | Abcam | Ab13970 | IF (1:500) |
| Antibody | anti-rabbit/mouse/chicken-Alexa-488 (GOAT) | Invitrogen | A-11008 rabbit A-1102 mouse A-1109 | IF (1:1000) |
| Antibody | anti-mouse/rat- Alexa-594 (GOAT) | Invitrogen | A-1102 mouse A-11007 rat | IF (1:1000) |
| Sequence-based reagent | gRNA | http://crispor.tefor.net/crispor.py | | CAATGTCCGAGTTACCACCA See Materials and methods: Generation of an EZR$^{-/-}$ HeLa cell line |
| Sequence-based reagent | hEZRNup | This paper | PCR primers | TGCCGTCGCCACACTGAGGA See Materials and methods: Generation of an EZR$^{-/-}$ HeLa cell line |
| Sequence-based reagent | hEZRNlow | This paper | PCR primers | TCCTTTGCTTCCATGCCTGG See Materials and methods: Generation of an EZR$^{-/-}$ HeLa cell line |
| Sequence-based reagent | olEzrin_Forward | This paper | PCR primers | GAACTCCTTCTAGCACCC See Materials and methods: Ezrin$^{-/-}$ medaka generation by CRISPR/Cas9 system |
| Sequence-based reagent | olEzrin_Reverse | This paper | PCR primers | CCGCCTCCCTCCTCAAATC See Materials and methods: Ezrin$^{-/-}$ medaka generation by CRISPR/Cas9 system |
| Sequence-based reagent | gRNA olEzrin Sense | This paper | | ACAATGGATGAGCCTATTAG See Materials and methods: Ezrin$^{-/-}$ medaka generation by CRISPR/Cas9 system |
| Sequence-based reagent | gRNA olEzrin antisense | This paper | | AGACTGATGCTGCCTCACTG See Materials and methods: Ezrin$^{-/-}$ medaka generation by CRISPR/Cas9 system |
| Peptide, recombinant protein | Dynabeads Protein G | Thermo Fisher Scientific | 10004D | Immunoprecipitation assay |
| Commercial assay or kit | iST Kit | Preomics | P.O.00027 | Peptides purification |
| Commercial assay or kit | mRNA sequencing library preparation of MEF | miRNeasy Micro Kit | 1071023 | |
| Commercial assay or kit | Cross-linking assay Lomant's reagent | Thermo Fisher | 22585 | |
| Commercial assay or kit | LysoTracker RED | Invitrogen | L7528 | |
| Commercial assay or kit | CellLight Early Endosomes- RFP | Invitrogen | C10587 | |
| Commercial assay or kit | Cathepsin B | Abcam | AB65300 | |
| Commercial assay or kit | Clarity Western ECL Substrate | Bio-Rad Laboratories | | |
| Chemical compound, drug | Cycloheximide (CHX) | Sigma-Aldrich | C4859 | Cell treatments |
| Chemical compound, drug | Bafilomycin A1 | Sigma-Aldrich | B1793 | Cell treatments |
| Chemical compound, drug | EGF | Peprotech | AF-100-15 | Cell treatments |
| Software, algorithm | MaxQuant | Andromeda search engine | | Mass spectrometry, all acquired raw files |
| Software, algorithm | ImageJ Software | *Schneider et al., 2012* | v. 1.54K | Immunofluorescence and western blot quantification |
| Software, algorithm | DAVID Bioinformatic tool | *Huang et al., 2009a*; *Huang et al., 2009b* | | Functional analysis on transcriptomics and proteomics data |
| Software, algorithm | iTEM software | Olympus SYS, Germany | | Fluorescence imaging |
| Software, algorithm | GraphPad Prism | Boston, Massachusetts USA. | 10.0.0 | Graphs |
| Commercial assay or kit | DAPI stain | Vector Laboratories | H-1200 | 1:500 |
| Commercial assay or kit | HBSS medium | Thermo Fisher Scientific | 14025092 | |
| Commercial assay or kit | HEPES | Thermo Fisher Scientific | 156330080 | |

## 3′ mRNA sequencing library preparation

The transcriptional response of four biological replicates for both MEF[WT] and MEF[Ezr] KO cell lines was analyzed using QuantSeq 3′ mRNA sequencing. RNA extraction, quality control, and preparation of RNA-seq libraries and sequencing on an NovaSeq6000 platform were carried out in collaboration with the Next Generation Sequencing (NGS) Facility at TIGEM following their standard procedures (*Carotenuto et al., 2022*). An average yield of ~4.5 Mb was obtained per sample.

## Computational analysis of deep sequencing data

Data analysis was performed using the pipeline already established at the Bioinformatics and Statistics Core Facility at TIGEM (*Pinelli et al., 2016*). Briefly, the reads were trimmed to remove adapter sequences and low-quality ends and reads mapping to contaminating sequences (e.g., ribosomal RNA, phIX control) were filtered out. Alignment was performed with STAR 2.6.0a3 (*Dobin et al., 2013*) on mm10 reference assembly obtained from cellRanger website4 (Ensembl assembly release 93). The expression levels of genes were determined with htseq-count 0.9.15 using mm10 Ensembl assembly (release 93) downloaded from the cellRanger website4. We filtered out all genes having <1 cpm in less than n_min samples and Perc MM reads >20% simultaneously. Differential expression analysis was performed using edgeR6 (*Liu et al., 2021*).

## Mass spectrometry

Protein extraction and preparation of MS samples were carried out in accordance with standard procedures currently utilized in the Mass Spectrometry Facility at TIGEM. About 30 mg of cell lysate was used. Peptides were purified using the iST Kit (Preomics) following the company instructions. Peptide separation and LC–MS/MS analysis were carried out accordingly to standard procedures as detailed in *Di Malta et al., 2023*.

## Data analysis of mass spectrometry

At least three independent biological replicates were performed for all experiments. For mass spectrometry, all acquired raw files were processed using MaxQuant (1.6.2.10) and the implemented Andromeda search engine. For protein assignment, spectra were correlated with the UniProt *Homo sapiens* including a list of common contaminants. Searches were performed with tryptic specifications and default settings for mass tolerances for MS and MS/MS spectra. Carbamidomethyl at cysteine residues was set as a fixed modification, while oxidations at methionine and acetylation at the N-terminus were defined as variable modifications. The minimal peptide length was set to seven amino acids, and the false discovery rate for proteins and peptide-spectrum matches to 1%. The match-between-run feature with a time window of 0.7 min was used. For further analysis, the Perseus software was used and first filtered for contaminants and reverse entries as well as proteins that were only identified by a modified peptide. For full proteomes and IP-interactomes, the LFQ Ratios were logarithmized, grouped and filtered for min. valid number (min. 3 in at least one group). Missing values were replaced by random numbers that are drawn from a normal distribution. Finally, the intensities were normalized by subtracting the median intensity of each sample. Significantly regulated proteins between conditions were determined by Student's *t*-test using FDR <0.05 as threshold.

## Functional analysis on transcriptomics and proteomics data

The threshold for the statistical significance of gene expression was FDR <0.05. The threshold for the statistical significance of the proteomics analysis was −log10 > 1.3 and −log2 > = 1. GOEA and KEGG pathway were performed on induced and inhibited genes, separately, both in the transcriptome and in the proteome experiments using the DAVID Bioinformatic tool (*Huang et al., 2009a*; *Huang et al., 2009b*) restricting the output to Biological Process, CC terms. The threshold for statistical significance of GOEA was FDR <0.1 and the Enrichment Score ≥1.5, while for the KEGG pathway analyses it was FDR <0.1. The comparison of the transcriptomics and proteomics identified 572 commonly regulated genes: 317 and 213 genes were induced and inhibited in both datasets, respectively.

## Data visualization

Heatmap and Venn diagram were generated using custom annotation scripts.

## Accession code

The transcriptomics data have been deposited in the NCBI Gene Expression Omnibus (GEO) (*Edgar et al., 2002*) and are accessible through GEO Series accession number GSE195983. The title of the dataset is: 'Transcriptome profile of EZR_KO cells'. For this dataset, a secure token has been created to allow a review of the record: private token. The proteome data were deposited in PRIDE repository and are available via ProteomeXchange with identifier PXD045157.

## Alpha fold

All protein pairs were modeled using AlphaFold3 (*Abramson et al., 2024*) through ChimeraX, using the alphafold dimers command to generate a JSON af3 query. The FASTA input for the command was downloaded from UniProt (https://www.uniprot.org). AlphaFold3 was run with default settings and random seed. The resulting structures were evaluated by analyzing the predicted Local Distance Difference Test scores, Predicted Aligned Error matrices, and protein interfaces with the alphafold interfaces command in ChimeraX. Simple bash/R scripts were used to mine associations between Pfam domains in the UniProt entries across various databases (PPIDM, 3DID, DOMINE). The domain interaction network was built in R (*R Development Core Team, 2024*) with tidyverse (https://doi.org/10.21105/joss.01686) and igraph (https://doi.org/10.5281/zenodo.7682609) and visualized using ggraph (https://CRAN.R-project.org/package=ggraph). All structural visualization/analyses were run with ChimeraX (*Pettersen et al., 2021*).

## Western blot analysis

After transfection and/or treatments, cells were collected to extract total protein, while mouse eyes were enucleated, and the retina was separated from the RPE. Both mice and cell samples were lysed using RIPA buffer (150 mM sodium chloride, 1% Triton X-100, 0.5% sodium deoxycholate, 0.1% sodium dodecyl sulfate, 50 mM Tris, pH 8.0) with an inhibitor cocktail (Thermo Fisher Scientific, 78420). The protein concentration was determined by Bradford analysis and quantified using a Thermo Fisher Helios γ spectrophotometer. Proteins were fractionated by sodium dodecyl sulfate–polyacrylamide gel electrophoresis and transferred to PVDF membranes (EMD Millipore, IPVH00010), then blocked in Tween 0.1% Tris-buffered saline containing 5% bovine serum albumin (BSA; Tocris 5217) for at least 1 hr at room temperature (RT) and subsequently incubated overnight at 4°C with primary antibodies. For western blot analysis, the following antibodies were used: mouse anti-NBR1 (1:1000, Abnova MO1), rabbit anti-LAMP1 (1:500, Sigma L1418), mouse anti-Ezrin (1:1000, Novex 357300), mouse anti-SQSTM1/P62 (1:1000, Abcam ab56416), rabbit anti-Cathepsin D (1:1000, Cell Signaling 2284), rabbit anti-LC3 (1:1000, Novus NB100-2220), mouse anti-GAPDH (1:1000, Santa Cruz SC-32233), rabbit anti-HER2/ErbB2 (1:1000, Cell Signaling 2165), rabbit anti-HER3/ErbB3 (1:1000, Cell Signaling 12708), rabbit anti-phospho-EGF receptor (Tyr845) (1:1000, Cell Signaling 6963), rabbit anti-EGF receptor (1:1000, Cell Signaling 4267), rabbit anti-MAPKAPK-2 (1:1000, Cell Signaling 3042), rabbit anti-phospho-MAPKAPK-2 (Thr222) (1:1000, Cell Signaling 3316), rabbit anti-p38 MAPK (1:1000, Cell Signaling 8690), rabbit anti-phospho-p38 MAPK (Thr180/Tyr182) (1:1000, Cell Signaling 4511), rabbit anti-ZO1 (1:1000, Abcam ab216880), mouse anti-EEA1 (1:1000, BD 610457), rabbit anti-Tuberin/TSC2 (1:1000, Cell Signaling 4308), rabbit anti-phospho-Tuberin/TSC2 (Ser939) (1:1000, Cell Signaling 3615), rabbit anti-phospho-Tuberin/TSC2 (Thr1462) (1:1000, Cell Signaling 3617), rabbit anti-p70 S6 Kinase (1:1000, Cell Signaling 9202), mouse anti-phospho-p70 S6 Kinase (Thr389) (1:1000, Cell Signaling 9206), rabbit anti-Akt (1:1000, Cell Signaling 9272), rabbit anti-phospho-Akt (Ser473) (1:1000, Cell Signaling 4060), rabbit anti-4E-BP1 (1:1000, Cell Signaling 9644), rabbit anti-phospho-4E-BP1 (Ser65) (1:1000, Cell Signaling 9456), rabbit anti-phospho-4E-BP1 (Thr37/46) (1:1000, Cell Signaling 2855), rabbit anti-Hamartin/TSC1 (1:1000, Cell Signaling 6935), mouse anti-EGFR (1:500, Santa Cruz sc-120), mouse anti-p-EGFR (1:500, Santa Cruz sc-57542). After washing three times with Tween 0.1% Tris-buffered saline (TBS-T), the membranes were incubated for 1 hr at RT with the following secondary antibodies: goat anti-rabbit IgG antibody, HPR conjugate, and goat anti-mouse IgG antibody HPR conjugate (1:10,000 EMD Millipore, 12-348; 12-349). Western blot detection was done with ChemiDoc XRS+ System-Bio-Rad and quantified using ImageJ software.

## Immunofluorescence

Mouse eyes were fixed overnight in 4% paraformaldehyde (PFA) in phosphate-buffered saline (PBS) at 4°C and then cryopreserved by treatment first with 5% and then with 30% sucrose in PBS and embedded in OCT (cryo embedding matrix). Twenty-micrometer cryosections were collected on slides (Superfrost Plus; Fisher Scientific, Pittsburgh, PA). Cells were fixed with 4% PFA (Chem Cruz sc-281692) for 15 min at RT followed by washing with 1% PBS. After fixation, the cells were permeated with blocking buffer (0.5% BSA, 0.005% saponin, 0.02% $NaN_3$) for 1 hr at RT. Medaka fish at stage 40 were subjected to anesthesia and then fixed by incubation in 4% PFA for 4 hr at RT. Samples were rinsed three times with PTW 1× (1× PBS, 0.1% Tween, pH 7.3) and then incubated overnight in 15% sucrose/PTW 1× at 4°C, and then again incubated overnight in 30% sucrose/PTW 1× at 4°C and embedded. Sixteen-micrometer cryosection were collected on slides. The following primary antibodies were used: rat anti-LAMP-1 (1:400, Santa Cruz sc-19992), mouse anti-LAMP-1 (1:1000, DSHB H4A3), rabbit anti-LAMP1 (1:100, Abcam ab24170), rabbit anti-LC3B (1:200, Novus NB100-2220), rabbit anti-EGF receptor (1:50, Cell Signaling 4267), mouse anti-EEA1 (1:100, BD 610457), rabbit anti-Tuberin/TSC2 (1:100, Cell Signaling 4308), rabbit anti-Hamartin/TSC1 (1:1000, Cell Signaling 6935), mouse anti-EGFR (1:50, Santa Cruz sc-120), chicken anti-GFP (1:500, Abcam ab13970), LysoTracker Red (Invitrogen L7528), CellLight Early Endosomes-RFP (Invitrogen C10587). All incubations were performed overnight at 4°C. After washing with 1% PBS, slides were incubated with the following secondary antibodies: Alexa 488 goat anti-rabbit/mouse/Chicken (1:1000, Invitrogen A-11008 rabbit, A-11032 mouse, A-11039), Alexa 594 goat anti-mouse/rat (1:1000, Invitrogen A-11032 mouse, A-11007 rat), and DAPI (1:500, Vector Laboratories H-1200) for 1 hr at RT; then, the slides were washed with 1% PBS and mounted with PBS/glycerol and imaged with a Zeiss LSM800 microscope. Three dimensional images were imaged with a Zeiss LSM880 confocal microscope equipped with Airyscan super-resolution imaging module, using ×63/1.40 NA Plan Apochromat Oil DIC M27 objective lens (Zeiss MicroImaging, Jena, Germany).

## Live cell imaging

HeLa cells were transiently transfected with EGFR-GFP and treated as indicated in the *Figure 5*. Time-lapse video was acquired for 5 min. One frame was acquired roughly every 0.5 s with lasers set at 30% power or below. TIRF time-lapse imaging was performed with a 60× Plan Apo oil immersion lens using a Nikon Eclipse Ti Spinning Disk microscope, and images were annotated, and the video was reconstitute using ImageJ software.

## Image analysis

### Lysotracker and endotracker quantification

Fluorescent images of the cells were captured at ×40 magnification using a LSM700 Zeiss Confocal Microscopy system, converted to gray-scale and normalized to background staining, using ImageJ. Quantification of lysotracker and endotracker reactivity was measured as mean values to define fluorescence signal intensity (IntDen/Area) and as the area occupied by fluorescent labeling in each region of interest.

### LC3–LAMP1, TSC1–LAMP1, and TSC2–LAMP1 co-localization

The co-localization of LC3 (green), LAMP1 (red), and TSC1/TSC2 (green) and LAMP1 (red) were evaluated using a LSM700 Zeiss Confocal Microscopy after immunostaining of endogenous proteins. Average values were calculated over 10 images, each containing a mean of 10 cells per image, and collected from at least three independent experiments. Exposure settings were unchanged throughout acquisition. Images were analyzed using the JaCoP plugin (*Bolte and Cordelières, 2006*) in ImageJ software.

### EGFR-positive endosome quantification

Morphometric analysis of the distribution of gold particles (EGFR-labeled) at endosomal structures was performed using iTEM software (Olympus SYS, Germany). In detail, we counted the number of EGFR-positive endosomes on almost 10 ×26,500 magnification images. In the absence of specific

staining, early endosome identification relied on morphological characteristics described in the literature (*Vogel et al., 2015*).

## Cathepsin B assay

Cathepsin B activity was measured by a fluorometric assay kit (AB65300; Abcam, Cambridge, MA, USA) following the manufacturer's instructions. The reaction and fluorescence were read at 400 nm (excitation) and 505 nm (emission) on Promega GloMax discover.

## Cross-linking assay

HeLa cells were washed twice with PBS and then cross-linked with DSP solution (Lomant's Reagent, Thermo Fisher 22585) at a final concentration of 1 mM for 30 min at RT. The reaction was stopped by adding stop solution (Tris-HCl 1 M, pH 7.5) at a final concentration of 10 mM for 15 min at RT. Each sample was analyzed by western blot assay.

## Immunoprecipitation assay

Cells were washed three times with ice-cold PBS and then homogenized with ice-cold lysis buffer (20 mM Tris-HCl pH 7.4, 150 mM NaCl, 1 mM EDTA, 0.5% NP40). At least 1 mg of proteins were immunoprecipitated with mouse anti-Ezrin (Novex, 357300) and rabbit anti-EGF receptor (Cell Signaling 4267) in rotation at 4°C overnight. Then, the immunoprecipitates were conjugated with protein G Beads (Dynabeads Protein G, Thermo Fisher Scientific 10004D), eluted in Laemmli buffer, and subjected to immunoblot analyses.

## Endosomal and membrane proteins extraction

For endosomal proteins, cultured cells ($1 \times 10^6$) were collected by low-speed centrifugation and washed with cold PBS. The pellet was resuspended in 500 µl of Buffer solution of Minute Endosome Isolation (Invent, biotechnologies, ED-028). The endosomal proteins extraction was performed in accordance with the manufacturer's instructions. For membrane proteins extraction, $5 \times 10^6$ cells were scraped off from plate surface and resuspended in growth media. After centrifuging the cells, the pellet was washed with Cell Wash Solution and then was resuspended in Permeabilization Buffer, accordingly with the manufacturer's instructions (Mem-PER Plus Membrane Protein Extraction Kit, Thermo Fisher, 89842).

## Cell culture and treatments

ARPE-19, HeLa, and MEF cell lines were obtained from American Type Culture Collection (ATCC). ARPE-19 cells were cultured in Dulbecco's modified Eagle medium (DMEM)/F-12, while HeLa and MEF cells were cultured in DMEM (Gibco) supplemented with 10% (vol/vol) FBS and 5% penicillin–streptomycin. All cell lines were maintained at 37°C, 5% $CO_2$ in a humified incubator according to the guidelines provided by the vendors. MEF EZR$^{KO}$ cells were kindly donated by Alessandra Eva of Istituto G. Gaslini, Genova, Italy. To analyze the autophagic flux, cells were treated with 200 nM Bafilomycin A1 (Sigma-Aldrich, B1793) for 3 hr in an incubator and maintained in starvation for 30 min in HBSS medium (Thermo Fisher Scientific, 14025092) supplemented with 10 mM HEPES (Thermo Fisher Scientific, 156330080). To evaluate EGFR localization in immunoelectron microscopy, HeLa cells were treated with 100 µg/ml cycloheximide (Sigma-Aldrich, C4859). Drug treatment was performed for 6 hr with 10 µM of NSC668394 or DMSO as previously reported (*Bulut et al., 2012*). EGF stimulation was obtained with 10 ng/ml of animal-free recombinant human EGF (Peprotech AF-100-15) for 3 hr. Insulin stimulation was achieved with 1 µM of Insulin solution human (Sigma I9278) for 30 min, after 16 hr of serum starvation. We used a sub-confluent cell culture (i.e., 80% of confluence) for each in vitro experiment.

## Generation of an EZR$^{-/-}$ HeLa cell line

HeLa (ATCC CCL-2) full knockout of the *EZRIN* gene was generated using the CRISPr/Cas9 system. The gRNA sequence CAATGTCCGAGTTACCACCA was selected using the http://crispor.tefor.net/crispor.py online tool. HeLa cells were electroporated using the Amaxa system with the nucleofection kit Cat No VCA-1003 from Lonza. Cells were FACS-sorted into 96-well plates to obtain single-cell-derived colonies carrying the INDEL mutations. Upon genomic DNA extraction, the genomic

sequence containing the targeted region was amplified by PCR reaction with the specific primers: hEZRNup TGCCGTCGCCACACTGAGGA, hEZRNlow TCCTTTGCTTCCATGCCTGG. PCR products were analyzed by DNA Sanger sequencing and the cell clone carrying the homozygous deletion *c.23 DEL AGTTACCACCATG* was selected and expanded.

## Plasmids and transfections

Cells were transfected at 80% confluence using Lipofectamine 2000 (Invitrogen, 12566014), following the manufacturer's protocol. The plasmids used were Ezrin$^{T567D}$ and Ezrin$^{T567A}$-mCherry, modified from vectors described by Coscoy et al., provided by the S.Coscoy lab (Insitute Curie, Paris) (*Coscoy et al., 2002*), EGFR-GFP (Addgene, 32751), TFEB-GFP (Addgene, 38119).

## Immunoelectron microscopy analysis

HeLa cells were fixed with a mixture of 4% PFA and 0.05% glutaraldehyde (GA) for 10 min at RT, then washed with 4% PFA once to remove the residual GA and fixed again with 4% PFA for 30 min at RT. Next, the cells were incubated with a blocking/permeabilizing mixture (0.5% BSA, 0.1% saponin, 50 mM NH$_4$Cl) for 30 min and subsequently with the primary monoclonal antibody anti-GFP, diluted 1:500 in blocking/permeabilizing solution. The following day, the cells were washed and incubated with the secondary antibody, an anti-rabbit Fab fragment coupled to 1.4 nm gold particles (diluted 1:50 in blocking/permeabilizing solution) for 2 hr at RT. The cells were then post-fixed as described in *Polishchuk and Polishchuk, 2019*. After dehydration, the specimens were embedded in epoxy resin and polymerized at 60°C for 72 hr. Thin 60 nm sections were cut on a Leica EM UC7 microtome. The EM images were acquired from thin sections using a FEI Tecnai-12 electron microscope equipped with a VELETTA CCD digital camera (FEI, Eindhoven, the Netherlands).

## RPE and retina dissection

To analyze protein expression levels in RPE individually, mouse eyes were dissected to remove optic nerve, cornea, lens, and retina in ice-cold PBS 1× under stereomicroscopy (Leica). The RPE was peeled from the eyecup and transferred to a tube containing 100 µl of RIPA buffer. RPE cells were pelleted by centrifugation at 12,000 × *g* for 15 min at 4°C.

## Light/dark adaptation of mice for tissue isolation

Mice were maintained in dark conditions with a maximum of 0.4 lux from 19:00 pm to 7:00 am. Then, animals were kept in a room with the light phase (450 lux) from 7:00 am to 19:00 pm. For light/dark transition studies, some animals were transferred after 3 hr from light conditions to dark conditions and sacrificed. Eyes from dark mice were isolated under dim red light.

## Medaka stocks

The cab strain of wild-type and Ezrin$^{-/-}$ medaka (*O. latipes*) lines were maintained following standard conditions (i.e., 12 hr/12 hr dark/light conditions at 27°C). Embryos were staged according to the method proposed by *Iwamatsu, 2004*. All studies on fish were conducted in strict accordance with the Institutional Guidelines for animal research and approved (n° 7B56B.0) by the Italian Ministry of Health, Department of Public Health, Animal Health, Nutrition and Food Safety in accordance with the law on animal experimentation (D. Lgs.26/2014). Furthermore, all fish treatments were reviewed and approved in advance by the Ethics Committee at the TIGEM institute (Pozzuoli (NA), Italy).

**Table 1.** Primer sequences.

| | Name | Sequence (5'–3') | Usage |
|---|---|---|---|
| | *olEzrin* Sense | ACAATGGATGAGCCTATTAG | |
| EZRIN-gRNA | *olEzrin* Antisense | AGACTGATGCTGCCTCACTG | CRISPR/Cas9-LoxP target site |
| Primers | *olEzrin*_Forward *olEzrin*_Reverse | GAACTCCTTCTAGCACCC CCGCCTCCCTCCTCAATC | PCR for olEzrin genotype |

## Ezrin⁻/⁻ medaka generation by CRISPR/Cas9 system

The genomic sequence of medaka *Ezrin* was obtained the medaka genome database at the Ensembl Genome Database Project (http://www.ensembl.org/Oryzias_latipes; ENSORLG00000012128). Design and construction of *OlEzrin*-sgRNA were committed to SYNTHEGO. The sequences of *OlEzrin*-sgRNA oligonucleotides are listed in *Table 1*. Instead of Cas9 mRNA, the commercial reagent of pCS2-nCas9n (Addgene, #4729) was used in this study. After pCS2-nCas9n was digested by NotI treatment, this linearized vector was used as the template for synthesizing capped Cas9 mRNA with a mMessage mMachine SP6 Kit (Life Technologies). Microinjection of the medaka embryos followed a method described preciously by *Kinoshita et al., 2000*. A mixture containing 200 ng/µl of Cas9 mRNA and 20 ng/µl of *OlEzrin*-sgRNA was prepared and injected into the fertilized eggs at the one-cell stage. After hatching, the larvae were raised to sexual maturity and used as 'founder' fish (F0). To observe the genomic DNA mutations induced by Cas9 and *OlEzrin*-sgRNA in CRISPR/Cas9-mediated Ezrin-mutated medaka, a small piece of the caudal fin from individual F0 fish was collected and subjected to genomic DNA analysis, using the primer set indicating in *Table 1*. After the above screening had confirmed the occurrence of CRISPR/Cas9-mediated Ezrin mutation in the F0 generation, these founder fish were crossed with each other, and their offspring (F1) were checked for Ezrin mutations in the same way. Two of the F1 progeny with the same mutation patterns were mated to produce the F2 generation. The F2 generation were crossed with each other to produce F3 progeny, which was screened as described above to confirm that the same mutation patterns were successfully inherited.

## Whole-mount immunostaining

Medaka larvae were fixed in 4% PFA, 2× PBS, and 0.1% Tween-20. The fixed larvae were washed with PTW 1× and digested for 20 min with 10 µg/ml proteinase K and washed twofold with 2 mg/ml glycine/PTW 1×. The samples were fixed for 20 min in 4% PFA, 2× PBS, and 0.1% Tween-20, washed with PTW 1×, and then incubated for 2 hr in FBS 1%/PTW 1×, at RT. The larvae were incubated with mouse anti-EGFR (1:50, Santa Cruz sc-120) overnight at 4°C. The samples were washed with PTW 1×, incubated with the secondary antibody, Alexa-488 goat anti-mouse IgG (Thermo Fisher), then with DAPI. Finally, the larvae were placed in glycerol 100%.

## Statistical analysis

### *T*-test, Welch's *t*-test, and Mann–Whitney test

For the analysis of the statistically significant differences between two conditions, we performed the Shapiro–Wilk test to check if each condition had followed the normal distribution (null hypothesis): we performed the non-parametric Mann–Whitney test in case of rejection of the null hypothesis (p-value <0.05), and we performed the parametric unpaired *t*-test in case of non-rejection of the null hypothesis (p-value ≥0.05). In the second case, we also performed the *F*-test to check the homoscedasticity between the compared conditions (null hypothesis): we applied the parametric Welch's *t*-test in case of rejection of the null hypothesis (p-value <0.05). All the tests were performed with GraphPad Prism 10.0.0, GraphPad Software, Boston, Massachusetts, USA.

### ANOVA, Welch's ANOVA, and Kruskal–Wallis test (with multiple comparisons post hoc tests)

For the analysis of the statistically significant differences among multiple conditions, we performed the Shapiro–Wilk test to check if each condition had followed the normal distribution (null hypothesis): we performed the non-parametric Kruskal–Wallis test in case of rejection of the null hypothesis (p-value <0.05), and we performed the parametric one-way ANOVA in case of non-rejection of the null hypothesis (p-value ≥0.05). In the second case, we also performed the Brown–Forsythe test to check the homoscedasticity between the compared conditions (null hypothesis): we applied the parametric Welch's one-way ANOVA in case of rejection of the null hypothesis (p-value <0.05). For completeness, we computed the p-values with post hoc tests for the pairwise multiple comparisons: Tukey's test for one-way ANOVA, Dunnett's test for Welch's one-way ANOVA, and Dunn's test for Kruskal–Wallis test. All the tests were performed with GraphPad Prism 10.0.0, GraphPad Software, Boston, Massachusetts, USA.

## Poisson regression

For the analysis of the statically significant differences between two conditions with discrete values (i.e., counts), we performed the Poisson regression over data, considering a generalized linear model with likelihood ratio test. No correction for multiple comparisons was necessary. Poisson regression with generalized linear model and likelihood ratio test were performed with the package 'car' (version 3.1-2) in the R environment (version 4.2.3).

# Acknowledgements

We are grateful to Dr. Cathal Wilson for critical reading and English editing of the manuscript. We are grateful to Edoardo Nusco for mice technical support. We also are grateful to Advanced Microscopy, Lysosomal Metabolism, Computational Biology and Medaka fish Cores at Department of Biology, University of studies of Naples Federico II. Acknowledgement is made to BioRender (https://www.biorender.com/) for model images. Work in the Conte group was supported by grants from the Million Dollar Bike Ride Grant Program MDBR-21-103-CHM, International Retinal Research Foundation, MIUR FISR2020IP_03551, MIUR PNRR_PRIN_P2022NPLZC, MIUR PRIN _2022WJFN5X, and Sanfilippo Children's Foundations and National MPS Society.

# Additional information

### Funding

| Funder | Grant reference number | Author |
|---|---|---|
| Orphan Disease Center, Perelman School of Medicine, University of Pennsylvania | MDBR-21-103-CHM | Ivan Conte |
| International Retinal Research Foundation | IRRF Grant | Ivan Conte |
| Ministero dell'Istruzione, dell'Università e della Ricerca | FISR2020IP_03551 | Ivan Conte |
| Ministero dell'Istruzione, dell'Università e della Ricerca | PNRR_PRIN_P2022NPLZC | Ivan Conte |
| Ministero dell'Istruzione, dell'Università e della Ricerca | MIUR PRIN _2022WJFN5X | Ivan Conte |
| Sanfilippo Children's Foundation | SCF GRANT | Ivan Conte |
| National MPS Society | MPS GRANT | Ivan Conte |

The funders had no role in study design, data collection and interpretation, or the decision to submit the work for publication.

### Author contributions

Giuliana Giamundo, Conceptualization, Data curation, Formal analysis, Validation, Investigation, Methodology, Writing – original draft, Writing – review and editing; Daniela Intartaglia, Conceptualization, Data curation, Formal analysis, Validation, Investigation, Methodology, Writing – original draft; Eugenio Del Prete, Elena Polishchuk, Fabrizio Andreone, Marzia Ognibene, Sara Buonocore, Bruno Hay Mele, Francesco Giuseppe Salierno, Jlenia Monfregola, Dario Antonini, Paolo Grumati, Alessandra Eva, Rossella De Cegli, Investigation; Ivan Conte, Conceptualization, Data curation, Formal analysis, Supervision, Funding acquisition, Validation, Investigation, Visualization, Methodology, Writing – original draft, Project administration, Writing – review and editing

## Author ORCIDs
Giuliana Giamundo ⓘ https://orcid.org/0000-0002-3101-4697
Eugenio Del Prete ⓘ https://orcid.org/0000-0003-3214-9021
Marzia Ognibene ⓘ https://orcid.org/0000-0003-3698-9319
Bruno Hay Mele ⓘ https://orcid.org/0000-0001-5579-183X
Paolo Grumati ⓘ https://orcid.org/0000-0002-9942-9389
Ivan Conte ⓘ https://orcid.org/0000-0002-8968-9021

## Ethics

All studies on animals were conducted in strict accordance with the institutional guidelines for animal research and approved by the Italian Ministry of Health, Department of Public Health, Animal Health, Nutrition and Food Safety in accordance with the law on animal experimentation article 31; D.L. 26/2014; protocol number: 0016304-21/07/2020-DGSAF-MDS-P.

Reviewer #2 (Public review): https://doi.org/10.7554/eLife.98523.3.sa1
Reviewer #3 (Public review): https://doi.org/10.7554/eLife.98523.3.sa2
Author response https://doi.org/10.7554/eLife.98523.3.sa3

---

# Additional files

## Supplementary files

Supplementary file 1. Comparison of the transcriptomics and proteomics analyses.

Supplementary file 2. Gene Ontology (GO) and differentially expressed genes (DEGs) on common genes.

Supplementary file 3. Cellular Compartment (CC) on common genes.

Supplementary file 4. Data from SILAC Phosphoproteomics, Kinase perturbations, and Proteomics drug atlas.

Supplementary file 5. Genes derived from GeneMANIA.

MDAR checklist

## Data availability

Sequencing data have been deposited in GEO Series accession number GSE195983. Proteome data was deposited in PRIDE repository and are available via ProteomeXchange with identifier PXD045157.

The following datasets were generated:

| Author(s) | Year | Dataset title | Dataset URL | Database and Identifier |
|---|---|---|---|---|
| De Cegli R | 2025 | Transcriptome profile of EZR_KO cells | https://www.ncbi.nlm.nih.gov/geo/query/acc.cgi?acc=GSE195983 | NCBI Gene Expression Omnibus, GSE195983 |
| De Cegli R | 2025 | Ezrin defines TSC1 activation at endosomal compartment through EGFR-AKT signaling | http://www.ebi.ac.uk/pride/archive/projects/PXD045157 | PRIDE, PXD045157 |

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
