## [Editor Report · eLife Assessment]

Giamundo et al. present **fundamental** data with new insights into the role of Ezrin, a major membrane-actin linker that assembles signaling complexes, in the spatial regulation of EGF signaling mediators. The use of multiple state-of-the-art microscopy techniques, multiple cell lines and inhibitors, together with in vivo models **convincingly** supports the majority of their conclusions. The findings are helpful for understanding EGF/mTOR signal transduction and support a critical role for the scaffolding protein Ezrin in the upstream regulation of EGFR/AKT activity, TSC subcellular localization and mTORC1 signaling. These findings contribute substantially to understanding how endo-lysosomal signaling are regulated, alterations which are implicated in many human diseases.

---

## [Referee Report · Reviewer #2 (Public review)]

Summary:

The authors begin with the stated goal of gaining insight into the known repression of autophagy by Ezrin, a major membrane-actin linker that assembles signaling complexes on membranes. RNA and protein expression analysis is consistent with upregulation of lysosomal proteins in Ezrin-deficient MEFs, which the authors confirm by immunostaining and western blotting for lysosomal markers. Expression analysis also implicates EGF signaling as being altered downstream of Ezrin loss, and the authors demonstrate that Ezrin promotes relocalization of EGFR from the plasma membrane to endosomes. Ezrin loss reduces downstream MAPK and Akt signaling, and represses mTORC1 signaling by promoting lysosomal localization of the TSC complex. An Ezrin mutant Medaka fish line is then generated to test its role in retinal cells, which are known to be sensitive to changes in autophagy regulation. Phenotypes in this model appear generally consistent with observations made in cultured cells, though milder overall.

Strengths:

Data on the impact of Ezrin-loss on relocalization of EGFR from the plasma membrane are extensive, and thoroughly demonstrate that Ezrin is required for EGFR internalization in response to EGF.

A new Ezrin-deficient in vivo model (Medaka fish) is generated.

Strong data demonstrating that Ezrin loss suppresses Akt signaling and mTORC1 signaling by promoting TSC complex localization to the lysosome.

Weaknesses:

The authors have addressed all concerns

---

## [Referee Report · Reviewer #3 (Public review)]

Summary:

In this study, the authors have attempted to demonstrate a critical role for the cytoskeletal scaffold protein Ezrin, in the upstream regulation of EGFR/AKT/MTOR signaling. They show that in the absence of Ezrin, ligand-induced EGFR trafficking and activation at the endosomes is perturbed, with decreased endosomal recruitment of the TSC complex, and a corresponding decrease in AKT/MTOR signaling.

Strengths:

The authors have used a combination of novel imaging techniques, as well as conventional proteomic and biochemical assays to substantiate their findings. The findings expand our understanding of the upstream regulators of the EGFR/AKT MTOR signaling and lysosomal biogenesis, appear to be conserved in multiple species, and may have important implications for the pathogenesis and treatment of diseases involving endo-lysosomal function, such as diabetes and cancer, as well as neuro-degenerative diseases like macular degeneration. Furthermore, pharmacological targeting of Ezrin could potentially be utilized in diseases with defective TFEB/TFE3 functions like LSDs. While a majority of the findings appear to support the hypotheses, there are substantial gaps in the findings that could be better addressed. Since Ezrin appears to directly regulate MTOR activity, the effects of Ezrin KO on MTOR-regulated, TFEB/TFE3 -driven lysosomal function should be explored more thoroughly. Similarly, a more convincing analysis of autophagic flux should be carried out. Additionally, many immunoblots lack key controls (Control IgG in CO-Ips) and many others merit repetition to either improve upon the quality of the existing data, validate the findings using orthogonal approaches or to provide a more rigorous quantitative assessment of the findings, as highlighted in the recommendation for authors.

Comments on revisions:

The authors have satisfactorily addressed most of the concerns raised in the prior version, and have significantly improved upon the overall findings in the revised version.

---

## [Author Response]

The following is the authors’ response to the original reviews.

**Reviewer #1 (Public Review):**
Summary:The authors demonstrate that, while the loss of Ezrin increases lysosomal biogenesis and function, its presence is required for the specific endocytosis of EGFR. Upon further investigation, the authors reveal that Ezrin is a crucial intermediary protein that links EGFR to AKT, leading to the phosphorylation and inhibition of TSC. TSC is a critical negative regulator of the mTORC1 complex, which is dysregulated in various diseases, making their findings a valuable addition to multiple fields of study. Their cell signaling findings are translatable to an in vivo Medaka fish model and suggest that Ezrin may play a crucial role in retinal degeneration.Strengths:Giamundo, Intartaglia, et al. utilized unbiased proteomic and transcriptomic screens in Ezrin KO cells to investigate the mechanistic function of Ezrin in lysosome and cell signaling pathways. The authors' findings are consistent with past literature demonstrating Ezrin's role in the EGFR and mTORC1 signaling pathways. They used several cell lines, small molecule inhibitors, and cellular and in vivo knockout models to validate signaling changes through biochemical and microscopy assays. Their use of multiple advanced microscopy techniques is also impressive.

We are grateful to the Editor and the Reviewers for their important and constructive comments, which amended us to improve our manuscript. We have now carried out new experiments and analyses to further support our findings.

Weaknesses:While the authors demonstrated activation of TSC1 (lysosomal accumulation) and inactivation of Akt (decreased phosphorylation in TSC1), as well as decreased mTORC1 signaling in Ezrin knockout cells, direct experiments showing the rescue of mTORC1 activity by AKT and TSC1 mutants are required to confirm the linear signaling pathway and establish Ezrin as a mediator of EGFR-AKTTSC1-mTORC1 signaling. Although the authors presented representative images from advanced microscopy techniques to support their claims, there is insufficient quantification of these experiments. Additionally, several immunoblots in the manuscript lack vital loading controls, such as input lanes for immunoprecipitations and loading controls for western blots.

We wish to thank the Reviewer for his/her important and constructive comments on our manuscript and to consider that our study provides new information for understanding the mechanism regulating TSC/mTORC1 pathway. We have now extensively revised the manuscript according to his/her suggestions. Indeed, to expand on the evidence demonstrating Ezrin as a mediator of EGFR-AKTTSC1-mTORC1 signaling, the revised manuscript includes quantification of all advanced microscopy images, rescue experiments demonstrating the role of Ezrin in AKT/TSC/mTORC1 molecular network, and controls for WBs and immunoprecipitations.

**Reviewer #2 (Public Review):**
Summary:The authors begin with the stated goal of gaining insight into the known repression of autophagy by Ezrin, a major membrane-actin linker that assembles signaling complexes on membranes. RNA and protein expression analysis is consistent with upregulation of lysosomal proteins in Ezrin-deficient MEFs, which the authors confirm by immunostaining and western blotting for lysosomal markers. Expression analysis also implicates EGF signaling as being altered downstream of Ezrin loss, and the authors demonstrate that Ezrin promotes relocalization of EGFR from the plasma membrane to endosomes. Ezrin loss impacts downstream MAPK/Akt/mTORC1 signaling, although the mechanistic links remain unclear. An Ezrin mutant Medaka fish line was then generated to test Ezrin's role in retinal cells, which are known to be sensitive to changes in autophagy regulation. Phenotypes in this model appear generally consistent with observations made in cultured cells, though mild overall.Strengths:Data on the impact of Ezrin-loss on relocalization of EGFR from the plasma membrane are extensive, and thoroughly demonstrate that Ezrin is required for EGFR internalization in response to EGF.A new Ezrin-deficient in vivo model (Medaka fish) is generated.Strong data demonstrates that Ezrin loss suppresses Akt signaling. Ezrin loss also clearly suppresses mTORC1 signaling in cell culture, although examination of mTORC1 activity is notably missing in Ezrin-deficient fish.

We thank the Reviewer for the recognition of our study and apologize for the insufficient evidence reported in the previous version of the manuscript. As requested by the Reviewer, we considerably expanded the number of experiments to support EZRIN/EGFR/TSC molecular network in regulating autophagy pathway in the revised manuscript. Furthermore, following the Reviewer’s comment we have expanded the interpretation of our findings in the "Discussion” section. We hope the new version of our manuscript will satisfy the Reviewer’s worries.

Weaknesses:LC3 is used as a readout of autophagy, however the lipidated/unlipidated LC3 ratio generally does not appear to change, thus there does not appear to be evidence that Ezrin loss is affecting autophagy in this study.

We certainly agree with the Reviewer on the importance of this issue and apologize for the lack of clarity. Ezrin is an already widely characterized protein participating autophagy pathway. Several studies, including our previous studies, demonstrated that both silencing and pharmacological inhibition of Ezrin may promote autophagy by promoting activation of TFEB, in part through the TRPML1-calcineurin signaling pathway (Naso et al 2020; Intartaglia et al 2022; Lou et al 2024). However, a full elucidation on how Ezrin controls autophagy is still not unknown. As suggested by the Reviewer, to reinforce our data, we have now fixed this inaccuracy by better elucidating this aspect in the revised manuscript. Accordingly, we have monitored the autophagic flux and LC3 expression level following the guidelines for the use and interpretation of assays for monitoring autophagy (4th edition) by Klionsky et al. 2021. The data presented in the new Figure supplement 1 now better support the notion that depletion of Ezrin increases autophagic flux. We hope the new version of our manuscript will satisfy the Reviewer’s worries.

The conclusion is drawn that Ezrin loss suppresses EGF signaling, however this is complicated by a strong increase in phosphorylation of the p38 MAPK substrate MK2. Without additional characterization of MAPK and Erk signaling, the effect of Ezrin loss remains unclear. Causative conclusions between effects on MAPK, Akt, and mTORC1 signaling are frequently drawn, but the data only demonstrate correlations. For example, many signaling pathways can activate mTORC1 including MAPK/Erk, thus reduced mTORC1 activity upon Ezrin-loss cannot currently be attributed to reduced Akt signaling. Similarly, other kinases can phosphorylate TSC2 at the sites examined here, so the conclusion cannot be drawn that Ezrin-loss causes a reduction in Akt-mediated TSC2 phosphorylation.

We agree with the Reviewer that this is an interesting and important question. However, we respectfully disagree with the Reviewer and feel that addressing this point by additional studies on both MAPK and ERK pathways, as the Reviewer suggests, is outside the scope of this manuscript. We therefore prefer to address these questions in future studies. However, following the Reviewer’s comment we have expanded the interpretation of our findings in the "Discussion” section. We hope the new version of our manuscript will satisfy the Reviewer’s worries.

In Figure 7, the conclusion cannot be drawn that retinal degeneration results from aberrant EGFR signaling.

We certainly agree with the Reviewer on the importance of this issue. We now fixed this inaccuracy by adding TUNEL staining that showed the retinal degeneration in Ezrin KO medaka fish. The results of these assays are described in the Results section and documented in revised Figure 7, panels H.

It is unclear why TSC1 is highlighted in the title, as there does not appear to be any specific regulation of TSC1 here.

We modified the title accordingly

In Figure 1 the conclusion is drawn that there is an increase in lysosome number with Ezrin KO, however it does not appear that the current analysis can distinguish an increased number from increased lysosome size or activity. Similarly, conclusions about increased lysosome "biogenesis" could instead reflect decreased turnover.

Following this Reviewer’s observation, we changed the text according to his/her suggestion.

Immunoprecipitation data for a role for Ezrin as a signaling scaffold appear minimal and seem to lack important controls.

We apologize for these inaccuracies. We have now carried out new experiments to further support our findings. Moreover, all blots were changed for better exposed images. In the revised Figures the controls were showed.

In Figure 3A it seems difficult to conclude that EGFR dimerization is reduced since the whole blot, including the background between lanes, is lighter on that side.

We now fixed this inaccuracy. The blots were changed for better exposed images in revised Figure 3, panel A. and quantified

In Figure 6C specificity controls for the TSC1 and TSC2 antibodies are not included but seem necessary since their localization patterns appear very different from each other in WT cells.

We apologize because we have created some confusion. We have now emended this mistake and revised all panels in Figure 6C (now Figure 6D) for consistency between figures and text. Concerning the specificity of TSC1 and TSC2 antibodies and staining, indeed, antibodies labelling was showing the ordinary pattern from TSC in the cells as stated in Menon et al. 2014. We would like to point out that the antibodies are the same indicated in Menon et al. 2014 and our data are not only based on TSC1 and TSC2 staining but on a considerable number of in vivo and in vitro experiments in which many and different markers were used by performing several complementary approaches (i.e. immunofluorescence, western blot analysis, Omics, etc.)

Menon S, Dibble CC, Talbott G, Hoxhaj G, Valvezan AJ, Takahashi H, Cantley LC, Manning BD. Spatial control of the TSC complex integrates insulin and nutrient regulation of mTORC1 at the lysosome. Cell. 2014 Feb 13;156(4):771-85.

In Figure 7 the signaling effects in Ezrin-deficient fish are mild compared to cultured cells, and effects on mTORC1 are not examined. Further data on the retinal cell phenotypes would strengthen the conclusions.

We thank the Reviewer for his/her comment. We have now fixed this inaccuracy in the revised manuscript. We added the analysis for p4EBP1 (S65), a mTORC1 substrate Figure 7 panel D.

In Figure 7F there appears to be more EGFR throughout the cell, so it is difficult to conclude that more EGFR at the PM in Ezrin-/- fish means reduced internalization.

We agree with the Reviewer that it is an important question that helped us to improve the quality of the data presented. As correctly noted by the Reviewer, EGFR protein level is increased due to EZRIN deletion. This is evident in Figure 7 panel F, in line with both proteomic analysis and in vitro experiments (Figure 2I; Figure 3E; Figure 5C). We also agree that the increase of EGFR protein level could strength the background of immunofluorescence. Therefore, to better represent the EGFR membrane translocation on flat mount RPE from medaka lines, we add a highlighting box showing it in both WT and KO medaka line in the revised Figure 7 panel F.

**Reviewer #3 (Public Review):**
Summary:In this study, the authors have attempted to demonstrate a critical role for the cytoskeletal scaffold protein Ezrin, in the upstream regulation of EGFR/AKT/MTOR signaling. They show that in the absence of Ezrin, ligand-induced EGFR trafficking and activation at the endosomes is perturbed, with decreased endosomal recruitment of the TSC complex, and a corresponding decrease in AKT/MTOR signaling.Strengths:The authors have used a combination of novel imaging techniques, as well as conventional proteomic and biochemical assays to substantiate their findings. The findings expand our understanding of the upstream regulators of the EGFR/AKT MTOR signaling and lysosomal biogenesis, appear to be conserved in multiple species, and may have important implications for the pathogenesis and treatment of diseases involving endo-lysosomal function, such as diabetes and cancer, as well as neuro-degenerative diseases like macular degeneration. Furthermore, pharmacological targeting of Ezrin could potentially be utilized in diseases with defective TFEB/TFE3 functions like LSDs. While a majority of the findings appear to support the hypotheses, there are substantial gaps in the findings that could be better addressed. Since Ezrin appears to directly regulate MTOR activity, the effects of Ezrin KO on MTOR-regulated, TFEB/TFE3 -driven lysosomal function should be explored more thoroughly. Similarly, a more convincing analysis of autophagic flux should be carried out. Additionally, many immunoblots lack key controls (Control IgG in co-IPs) and many others merit repetition to either improve upon the quality of the existing data, validate the findings using orthogonal approaches, or provide a more rigorous quantitative assessment of the findings, as highlighted in the recommendation for authors.

We thank the Reviewer for the recognition of our study and apologize for the inaccuracies previously. We also greatly appreciate the efforts the reviewer went through with his/her support and help for the improvement of our manuscript. We considerably expanded the number of experiments to support EZRIN/EGFR/AKT network in controlling mTORC1 pathway in the revised manuscript as requested by the Reviewer. We hope the new version of our manuscript will satisfy the Reviewer’s worries.

**Reviewer #1 (Recommendations for The Authors):**
Major comments:(1) While the authors show that, in the absence of Ezrin, TSC accumulates on the lysosome and suppresses mTORC1 signaling, they should perform additional genetic experiments to strengthen their conclusions. Can they knockout or knockdown TSC1/2 in Ezrin-deficient cells to rescue mTORC1 activity? Can they mutate the lysosomal localization signal on TSC1 (TSC1Q149E/R204E/K238E) in Ezrin-deficient cells to rescue mTORC1 activity? Does constitutively active AKT (myr-AKT or AKT-E40K) restore mTORC1 activity in Ezrin-deficient cells?

We agree with the Reviewer that it is an important concern that helped us to improve the quality of the data presented. We now provide in the revised version of Figure supplement 4F the results of pharmacological inhibition of Ezrin on MEF-TSC2 KO cells. In line with our findings, the lack of TSC2 is able to rescue mTORC1 signaling in absence of Ezrin activity. Thus, these data strongly support that Ezrin is required for TORC1pathway via TSC complex targeting.

(2) In the absence of Ezrin, TSC1 constitutively localizes on the lysosome and suppresses mTORC1. Does this suppression hold in the presence of other mTORC1-activating signals (i.e., amino acids, insulin, oxygen)?

Following the reviewer’s suggestion we now provide this information in the revised Figure 6C, in which we showed that stimulation with insulin does not exert its activating effect on mTORC1 signaling (i.e. phosphorylation of pP70 S6 - pT389). These new data, together with the experiments on MEF TSC2 KO cells, clearly support the model by which Ezrin works as a scaffold protein connecting ATK signaling to TSC complex. The lack of Ezrin induces a disconnection between AKT and TSC complex, which is translocated on lysosomes and insensitive to inhibition of AKT signaling.

(3) In Figure 3A, the authors showed EGFR dimerization through a western blot of a crosslinking assay. However, the western blot data are unclear and do not strongly support their statement. Additionally, the authors mentioned that the dimerization is confirmed by immunofluorescence analysis, but this statement should be revised since the imaging analysis only indirectly shows the copresence of EZR and EGFR, not necessarily the dimerized EGFR. The authors should perform additional experiments to strengthen their claim or tone down their statements in the text and model figure.

We certainly agree with the Reviewer on the importance of this issue and now we have fixed this inaccuracy in the revised manuscript. The blots of crosslinking were changed for better exposed images in revised Figure 3, panel A. Moreover, we also properly quantified signals to support our conclusion.

(4) It is interesting that Ezrin binds EGFR, AKT, and TSC as a scaffolding protein. To define the mechanisms by which Ezrin interacts with AKT, EGFR, and TSC, can the authors perform domain analyses to determine which regions of Ezrin are required for its binding with AKT, EGFR, and TSC in mediating EGFR-AKT-TSC-mTORC1 signaling?

We thank the Reviewer for his/her comment that improves our manuscript. Conducting domain analysis in the lab would be ideal, although this seems to us a long tour de force that might be associated to several technical and experimental issues. However, in silico approaches provide a helpful alternative for generating initial hypotheses about domain-domain interactions, though they should be seen as a starting point rather than a complete solution. Recent advances in fold prediction suggest that AlphaFold3 could be used to predict dimer formation and, consequently, domain-domain interactions. However, such an approach is challenging in this case because some of the considered proteins are transmembrane, and all are prone to form multimeric complexes with multiple partners, making them poor candidates for reliable fold predictions. In fact, the predicted dimers are poorly supported, and AlphaFold3 lacks confidence in the relative positioning of interactors, limiting its interpretability. Alternatively, database mining and machine-learning methods, such as HINT, Domine, and PPIDomainMiner, provide more robust evidence. Indeed, these tools allow us to consistently identify a strong interaction between Ezrin's FERM central domain and EGFR's PK domain shown now in the Figure Supplement 2C and Supplement Figure 3C-H. Importantly, these findings generate valuable hypotheses, therefore experimental validation is still necessary. But we prefer to leave it for future studies.

Minor Comments:(1) There are several immunoblots that did not have adequate controls: - In Figure 2D, an input lane should be shown for each of the cell lysates to demonstrate the presence of other proteins in the cell lysate used for the IP.

We have now fixed this inaccuracy in the revised manuscript.

- Figure 3A does not have a loading control. Also, immunoblot quality should be significantly improved.

We have now fixed this inaccuracy in the revised manuscript.

- The HER2 western blot in Figure 5C does not accurately represent the data shown in the quantification graph.

We have now fixed this inaccuracy by replacing HER2 western blot in the revised Figure 5C.

- In Figure 6A, the authors should include an input as a control for the IP. To further support their claim in the model figure, can the authors also probe the IP lysate for Ezrin and Tsc2? If all are indeed in a complex together, they should be present.

Following this Reviewer’s observation, we add the input as control in the IP in the revised Figure 6A. Moreover, we include the immunoprecipitation data for the EZRIN and TSC2 interaction, accordingly (Figure 6A).

- Phosphorylation sites across figures should be uniformly annotated for consistency and ease of understanding, e.g., pTSC2(S939), pS6K1(T389), and pAKT(S473).

We have now fixed this inaccuracy in the revised text.

(2) There are several microscopy data that lack adequate quantification. For instance, Figures 2E, 2F, 3C, 4A, 5A, and 6F only show very few cells as representative images, which is not sufficient to support their claims.

We thank the Reviewer for his/her comment that improves our manuscript. Accordingly, we add adequate quantification and statistical analysis in the revised Figures, accordingly.

(3) Some suggestions to improve the readability of the manuscript:- In the abstract (line 32): "Loss of Ezrin was deficient in TSC repression by EGF and culminated in translocation of TSC to lysosomes triggering suppression of mTORC1 signaling." The wording is somewhat confusing, please change such as "Loss of Ezrin was not sufficient to repress TSC by EGF and culminated..." or "Loss of Ezrin blunted EGF-induced TSC suppression and culminated..."

We apologize for the lack of clarity and now we have fixed this inaccuracy by better elucidating this aspect in the revised manuscript.

- Figure 3D has a typo in the western blot labeling. Please change Citosol to Cytosol.

We have now fixed this inaccuracy in the revised text.

- Line 291: "Moreover, TSC2 resulted activated and AKT/mTOR signaling..." The wording is confusing.

We have now fixed this inaccuracy in the revised text. The text now reads: “Moreover, we found that TSC2 was dephosphorylated in response to light in the retina, when inactive Ezrin (Naso et al., 2020) and EGFR are weakly expressed (Figure supplement 6C as a consequence of a decrease of the AKT/mTORC1 signaling…..)

- The model in Figure 8 indicates that upon EGF stimulation, the activated Ezrin interacts with EGFR, causing its dissociation from actin filaments and leading to its endosome incorporation. However, the authors did not provide supporting data for this claim. Can the authors either cite literature or provide data for this? Otherwise, the model should be edited to remove actin filaments in the model.

We have now fixed this inaccuracy by removing actin filaments in the revised model.

**Reviewer #2 (Recommendations For The Authors):**
The data and written text seem to deal entirely with mTORC1, rather than mTORC2, thus it seems "mTOR" should be changed to "mTORC1" throughout.

We have now fixed this inaccuracy in the revised manuscript.

For clarification, the TSC protein complex should be referred to as the "TSC complex", whereas "TSC" generally refers to the tumor syndrome Tuberous Sclerosis Complex.

We have now fixed this inaccuracy in the revised manuscript.

Quantification of colocalization would be helpful in all the panels where it is currently missing.

We thank the Reviewer for his/her comment that improves our manuscript. Accordingly, we add adequate quantification of colocalization for each immunofluorescence in the revised Figures, accordingly.

Line 84 typo "thorough" should be "through"

We have now fixed this inaccuracy in the revised manuscript.

Line 178 - typo

We have now fixed this inaccuracy in the revised manuscript.

Line 209 - typo

We have now fixed this inaccuracy in the revised manuscript.

**Reviewer #3 (Recommendations For The Authors):**
Fig. 1 The data showing an increase in lysosomal biogenesis suggests an increase in transcriptional activity. This should be confirmed by one or more of the following: (1) Increased TFEB/TFE3 nuclear localization following EZR loss, (2) Increased CLEAR promoter luciferase activity assays, (3) Increased expression of multiple CLEAR transcripts (https://www.science.org/doi/10.1126/science.1174447) or (4) Increased TFEB/ TFE3/ CLEAR gene signatures by RNA seq. Similarly, data showing increased autophagic flux should be confirmed in the presence of chloroquine or bafilomycin.

We agree with the Reviewer that it is an important concern that helped us to improve the quality of the data presented. It is well established that a major mechanism regulating TFEB activity is represented by the nuclear translocation. We have now carried out new experiments demonstrating that depletion of Ezrin induces TFEB nuclear translocation in Ezrin^-/-^ cells. These findings are in line with our previous data in which pharmacological inhibition and silencing of Ezrin induced the same cellular phenotype. We also apologize because we have created some confusion, because we already carried out experiments with Bafilomycin to confirm the increase of autophagic flux. Therefore, the blots of autophagic flux were changed for better exposed images in revised Figure supplement 1H and the text was modified to emphasize these findings, accordingly.

Fig 2D, the lanes with EZR -/- cells expressing the EZR mutants should be repeated on the same gel as the first 2 lanes (with the WT and EZR^-/-^ cells)

We thank the Reviewer for his/her comment that improves our manuscript. In order to avoid any confusion, when describing the results in Figure 2D, we have now modified the Figure 2D, providing the required controls in the response to Reviewer #1 and #2. We hope the new version of our data will satisfy the Reviewer’s worries.

Fig 2F- The presence of reduced EGFR in intracellular compartments in Ezrin KO/ -/- cells should be quantified, and shown for a 2nd EZR null cell line as well (Ezrin null MEFs)

We added EGFR quantification in Figure 2F. We have now carried out new experiments demonstrating that EGFR is localized on cytoplasmic membrane in MEF Ezrin KO (Figure supplement 2H), accordingly.

Fig 2G, did the authors test the effects of EZR depletion on basal and EGF stimulated EGFR autophosphorylation on Y1068 and Y1045 as well as downstream activation of p42/44 ERK MAPK? Those should be tested in the HeLa system as well as the MEFs cells with EZR KO.

Following the Reviewer’s request, we have now added western blot data for EGFR autophosphorylation on Y1068 and p42/44 ERK MAPK in Figure 5C. Moreover, we have now added western blot data for p42/44 ERK MAPK on MEF cells in Figure supplement 2F. In contrast, we cannot provide any data for EGFR autophosphorylation on Y1068, because the antibody was not working on proteins from MEF cells.

Also, why would HER3 levels be expected to decrease? There seems to be minimal change in HER3 expression. Also, the significance of increased MK2 phosphorylation should be further elaborated.

The Reviewer raised justified concerns about the HER3 and MK2. We have discussed these aspects in the "results section”, accordingly.

Fig 3A- Crosslinking of EGFR is not very apparent in this blot. The crosslinking blots should be repeated 3 times and quantified.

We certainly agree with the Reviewer on the importance of this issue and now we have fixed this inaccuracy in the revised manuscript. The blots of crosslinking were changed for better exposed images in revised Figure 3, panel A. Moreover, we also properly quantified signals to support our conclusion.

Fig 3D- How were membrane endosomes isolated? This should be stated in the methods. Membrane/ Cytosol and Endosome fractionation showing EGFR levels should be shown in Ezrin null MEFs as well, and membrane expression should be further substantiated with surface biotinylation for cell surface EGFR.

We now report more information about the method that we used for membrane endosomes isolation in the Materials and Methods section. Following the Reviewer’s request, we also show that EGFR was not localized on endosomes upon EGF on Ezrin null MEFs. This data was reported in the new revised Figure Supplement 2G. Moreover, we have now carried out new experiments demonstrating the membrane localization of EGFR in MEF Ezrin KO cells. These findings are shown in Figure supplement 2H.

Fig 5C: Similar to 2G, EGFR autophosphorylation on Y1068 and Y1045 should also be measured, as well as downstream activation of p42/44 ERK MAPK?

Following the Reviewer’s request, we have now carried out new experiments to assess the EGFR autophosphorylation on Y1068 and Y1045, as well as downstream activation of p42/44 ERK MAPK. We added these new data in the revised Figure 5C, accordingly.

Fig 5D: Similar to 3D, Membrane/ Cytosol and Endosome fractionation showing EGFR levels should be shown in Ezrin null MEFs as well, and further substantiated with surface biotinylation for cell surface EGFR.

Following the Reviewer’s request, we show that EGFR was not localized on endosomes upon EGF (Figure Supplement 2G).

Supplement 2E: The blots show lower expression of EGFR and higher MAPK activation in EZR KO cells, contradicting the data in the other cells.

We apologize because we have created some confusion. It occurred during the preparation of Figure supplement 2E, reflecting image of a previous not finalized version of the Figure. We have now removed the error and replaced with a correct WB panel.

Supplement 2F: The authors should repeat the NSC668394 experiment using: (1) multiple doses, (2) In both the Ezrin KO and null cell lines (3) and repeat 3X to quantify differences in total EGFR.

We respectfully disagree with the Reviewer and feel that addressing this point by additional studies on dose response of NSC668394, as the Reviewer suggests, is outside the scope of this manuscript. However, we would like to point out that we have already conducted extensive studies on the doseresponse effects of NSC668394 administration in vitro (Patent: WO2020070333A1).

Moreover, we apologize for not having provided enough information about the number of biological independent replicates for WB analyses. Therefore, to fill this gap of information we have expanded the Material and Methods section, accordingly.

Patent: WO2020070333A1 - Ezrin inhibitors and uses thereof

Fig 6A: The IP experiments should be repeated with Control IgG

We have now fixed this inaccuracy in the revised manuscript.

Typos:(1) Figure 3D: Citosol

We have now fixed this inaccuracy in the revised manuscript.

(2) Line 216-217: "increased EGFR protein 217 levels on purified membranes and endosomes (Figure 3D and E)" - That should be decreased EGFR on endosomes in accordance with Figure 3D (lower panels)

We have now fixed this inaccuracy in the revised manuscript.

(3) Abstract: "Consistently, Medaka fish deficient for Ezrin exhibit defective endo-lysosomal pathway"

We have now fixed this inaccuracy in the revised manuscript.